

# Assimilation of 3D Polarimetric Microphysical Retrievals in a Convective-Scale NWP System

Lucas Reimann[1], Clemens Simmer[1], Silke Trömel[1,2]

[1]Department of Meteorology, Institute for Geoscience, University of Bonn, Bonn, 53121, Germany

[2]Laboratory for Clouds and Precipitation Exploration, Geoverbund ABC/J, Bonn, 53121, Germany

Correspondence to: Lucas Reimann (lreiman1@uni-bonn.de)

**Abstract.** This study assimilates for the first time polarimetric C-band radar observations from the German meteorological service (DWD) into DWD's convective-scale model ICON-D2 using DWD's ensemble-based KENDA assimilation framework. We compare the assimilation of conventional observations (CNV) with the additional assimilation of radar reflectivity Z (CNV+Z), with the additional assimilation of liquid or ice water content (LWC or IWC) estimates below or above the melting layer instead of Z where available (CNV+LWC/Z or CNV+IWC/Z, respectively). Hourly quantitative precipitation forecasts (QPF) are evaluated for two stratiform and one convective rainfall event in the summers of 2017 and 2021.

With optimized data assimilation settings (e.g., observation errors), the assimilation of LWC mostly improves first guess QPF compared to the assimilation of Z alone (CNV+Z), while the assimilation of IWC does not, especially for convective cases, probably because of the lower quality of the IWC retrieval in these situations. Improvements are, however, notable for stratiform rainfall in 2021, for which the IWC estimator profits from better specific differential phase estimates due to a higher radial radar resolution compared to the other cases. The assimilation of all radar data sets together (CNV+LWC+IWC+Z) yields the best first guesses.

All assimilation configurations with radar information consistently improve deterministic nine-hour QPF compared to the assimilation of only conventional data (CNV). Forecasts based on the assimilation of LWC and IWC retrievals on average slightly improve FSS and FBI compared to the assimilation of Z alone (CNV+Z), especially when LWC is assimilated for the 2017 convective case and when IWC is assimilated for the high-resolution 2021 stratiform case. However, IWC assimilation again degrades forecast FSS for the convective cases. Forecasts initiated using all radar data sets together (CNV+LWC+IWC+Z) yield the best FSS. The development of IWC retrievals more adequate for convection constitutes one next step to further improve the exploitation of ice microphysical retrievals for radar data assimilation.



## 1 Introduction

Heavy precipitation events can pose serious risks to the public and have increased in
frequency and strength since the middle of the 20[th] century (IPCC, 2021). Thus, improving
quantitative precipitation forecasts (QPF) is and remains of high societal interest. With the
ever-increasing computing power of meteorological forecasting centers, the resolution of
operational numerical weather prediction (NWP) models has increased up to the convective
scale, allowing more accurate QPF. NWP requires model states close to the true atmospheric
state (model initialization), which is usually achieved by combining short-term model forecasts
(first guesses) and observational data statistically, taking into account their respective
uncertainties, a process known as data assimilation (DA; e.g., Talagrand, 1997). Proper
initialization at convective scales is challenging, because uncertainties in convective processes
are difficult to estimate, and because of the observations required to resolve moist convective
processes. Weather surveillance radars can provide such data with unique temporal and
spatial resolution, and have become an indispensable data source for convective-scale NWP
over the past decades.

Radar observations have been successfully assimilated into convective scale NWP models,
e.g. with 4D variational (4DVar; e.g., Lewis and Derber, 1985; Le Dimet and Talagrand, 1986)
and 3D variational (3DVar; Courtier et al., 1998) DA methods (e.g., Sun and Crook, 1997,
1998; Xiao et al., 2005). Over the past two decades, radar DA using the ensemble Kalman
filter (EnKF; Evensen, 1994), a Monte Carlo approximation of the original Kalman filter
(Kalman, 1960), has become increasingly popular particularly due to its ability to estimate the
flow-dependent forecast uncertainty (the error covariance matrix) at the convective-scale
through an ensemble of model forecasts (e.g., Snyder and Zhang, 2003; Tong and Xue, 2005;
Aksoy et al., 2009; Dowell et al., 2011; Tanamachi et al., 2013; Wheatley et al., 2015; Bick et
al., 2016; Gastaldo et al., 2021). However, running a forecast ensemble of sufficient size to
robustly estimate the forecast error covariance matrix is not feasible in operational routines
due to the connected high computational costs, which can lead to sampling errors that can
cause filter divergence and spurious long-range correlations in the model domain (e.g.,
Houtekamer and Mitchell, 1998; Hamill et al., 2001). Observation localization (Ott et al., 2004),
which limits the radius within which observations affect the analysis, is a common approach to
mitigate this problem. The Local Ensemble Transform Kalman Filter (LETKF; Hunt et al., 2007),
a manifestation of the EnKF in which observation localization is a key feature and which
computes analyses at each grid point independently allowing for easy parallelization, is
currently very popular in the DA community. In addition to being used for research purposes
at the Japan Meteorological Agency (e.g., Miyoshi et al., 2010) and the European Centre for
Medium-Range Weather Forecasts (e.g., Hamrud et al., 2015), the LETKF has been
implemented operationally at the Italian Operational Centre for Meteorology (Bonavita et al.,



2010) as well as at the German Meteorological Service (Deutscher Wetterdienst, DWD), and
MeteoSwiss. Assimilation of 3D radar observations with the LETKF has shown positive effects
on short-term QPF (e.g., Bick et al., 2016; Gastaldo et al., 2021); at DWD, 3D radar DA with
the LETKF became operational for the convective-scale NWP model ICON-D2 (limited area
setup of the Icosahedral Nonhydrostatic model over Germany; Zängl et al., 2015) in spring
75  2021.

Radar DA has mainly focused on the horizontal radar reflectivity factor (hereafter simply
reflectivity) Z and the radial velocity V, with only Z providing direct information on cloud and
precipitation microphysical processes. Dual-polarization (i.e., linear orthogonal polarization
diversity; Seliga and Bringi, 1976, 1978; hereafter referred to as polarimetric) radar
observations provide additional information on clouds and precipitation, such as the size,
shape, orientation, and composition of hydrometeors (e.g., Zrnic and Ryzhkov, 1999).
Therefore, polarimetric radar observations can help to improve the representation of cloud-
precipitation microphysics in NWP models, weather analyses, and consequently short-term
QPF through model evaluation, parameterization developments, and DA (e.g., Kumjian, 2013;
Zhang et al., 2019). Polarimetric radar observations have already been used to improve
attenuation correction (e.g., Bringi et al., 1990; Testud et al., 2000; Snyder et al., 2010),
quantitative precipitation estimation (e.g., Zrnic and Ryzhkov, 1996; Ryzhkov et al., 2005a;
Tabary et al., 2011; Chen et al., 2021), severe weather observation and detection (e.g.,
Ryzhkov et al., 2005b; Bodine et al., 2013), hydrometeor classification (e.g., Park et al., 2009),
and model evaluation (e.g., Jung et al., 2012; Putnam et al., 2014, 2017). However, exploitation
of polarimetric information in DA is still in its infancy. One reason is the remaining uncertainties
in the relationships between polarimetric radar moments and model microphysical state
variables. Another reason is the lack of widespread operational polarimetric radar observations
from national surveillance radar networks in the past. In recent years, many of these networks
have been upgraded to polarimetry, e.g., in Germany, the USA, Canada, the UK, and China,
providing a valuable new source of observational data for future operational NWP.

Polarimetric moments can be linked to microphysical model state variables using either radar
forward operators or retrieval algorithms. Radar forward operators compute synthetic radar
moments based, e.g., on simulated parameterized particle size distributions, while retrievals
estimate microphysical model state variables from radar observations prior to DA. The direct
approach via forward operators is challenging because, e.g., hydrometeor shape, size, and
orientation distributions, all of which affect (polarimetric) radar observations, are still rather
rudimentarily represented in NWP models (e.g., Schinagl et al., 2019). The indirect approach
via retrievals circumvents these model deficiencies, but suffers from retrieval uncertainties. A
few case studies from the USA, Japan, and China have already attempted the direct DA of
polarimetric observations with some success using the EnKF (e.g., Jung et al., 2008; Jung et



al., 2010; Putnam et al., 2019; Zhu et al., 2020; Putnam et al., 2021) or the 3DVar method (e.g., Li et al., 2017; Du et al., 2021). Other studies have assimilated polarimetric observations indirectly via retrieved hydrometeor mixing ratios using the 4DVar approach (e.g., Wu et al.,
2000), the 3DVar method (e.g., Li and Mecikalski, 2010, 2012), or the EnKF method (e.g., Yokota et al., 2016). Polarimetric data have also been used to modify cloud analysis schemes based on polarimetric signatures in storms (Carlin et al., 2017) or to improve hydrometeor classifications (Ding et al., 2022). To our knowledge, no study has yet assimilated polarimetric radar data in Central Europe. In preparation for the direct assimilation of polarimetric data, the
single-polarization radar forward operator EMVORADO (Efficient Modular Volume Scanning Radar Forward Operator; Zeng et al., 2016), used operationally at DWD for the ICON-D2 model, is currently being upgraded to polarimetric capabilities, but is still in a testing phase. Regarding indirect assimilation, polarimetric retrieval algorithms for liquid and ice water content (LWC and IWC) have been proposed in the literature (e.g., Ryzhkov et al., 1998; Bringi and
Chandrasekar, 2001; Doviak and Zrnic, 2006; Carlin et al., 2016; Ryzhkov and Zrnic, 2019; Bukovcic et al., 2020; Carlin et al., 2021), but most of these algorithms were developed with a focus on S-band radars in the USA. The applicability of these retrieval relations for Germany with its C-band radar network and its quite different precipitation climatology may thus be limited. Recently, a hybrid polarimetric LWC estimator adapted to the German national C-band
network has been developed by Reimann et al. (2021).

The present paper takes a first step towards the indirect assimilation of polarimetric radar observations using microphysical retrievals of LWC and IWC in Germany and evaluates their impact on short-term QPF relative to the direct assimilation of Z observations. Polarimetric radar observations from the German national C-band weather radar network are assimilated
into the DWD ICON-D2 model using the corresponding DA framework KENDA (Kilometre-scale Ensemble Data Assimilation; Schraff et al., 2016) implementing the LETKF scheme. LWC and IWC data are estimated from the polarimetric measurements below and above the melting layer using the hybrid retrievals of Reimann et al. (2021) and Carlin et al. (2021), respectively. We attempt to identify suitable assimilation configurations for LWC and IWC
based on first-guess QPF quality and provide first insights into how the indirect assimilation of polarimetric information affects short-term QPF up to nine hours lead time. The study focusses on three intense precipitation periods in the summers of 2017 and 2021 over Germany.

The remainder of the paper is structured as follows. Section 2 briefly introduces the ICON-D2 model and the KENDA DA framework. Section 3 describes the data used and the applied
microphysical retrieval algorithms. Section 4 shows the experimental setup including the technique of assimilating the LWC and IWC retrievals and the experiment parts. Section 5 presents the results of the experiments, and Sect. 6 presents the conclusions.



## 2 Forecast model and assimilation framework

### 2.1 The ICON-D2 model

The ICON (Icosahedral Nonhydrostatic) modelling framework (Zängl et al., 2015) is a global NWP and climate modelling system jointly developed by DWD and the Max Planck Institute for Meteorology in Hamburg, Germany, and became operational in DWD's forecasting system in 2015. In this study, we perform integrations with the convection-resolving, area-limited setup of the ICON model, ICON-D2, covering Germany and parts of its neighboring states. The

ICON-D2 model uses an unstructured triangular grid with a resolution of about 2.2 km horizontally and 65 vertical levels; the near-ground levels are terrain-following and the higher levels gradually shift to constant heights towards the model top. Lateral boundary conditions are provided by simulations of the ICON-EU model, a nesting setup of the global ICON model over Europe. The ICON-D2 model became operational at DWD recently, ousting the previously

used COSMO (Consortium for Small-scale Modelling) model (Baldauf et al., 2011).

The ICON-D2 model provides prognostic variables including the 3D wind velocity components and the virtual potential temperature. The total density of the air-water mixture and the individual mass fractions of dry air, water vapor, cloud water, cloud ice, rain, snow, and graupel are further prognostic variables, which are simulated in our study with the single-moment

microphysics scheme representing a two-component system of dry air and water, which can occur in all three states of matter.

### 2.2 The KENDA framework

The KENDA system, originally developed for the COSMO model, is now operationally used for the ICON-D2 model at DWD and includes the LETKF scheme (see Appendix A or Hunt et al.

(2007) for more details on the LETKF). KENDA employs one deterministic model run in addition to the current 40-member ensemble (40+1-mode), which is updated in the analysis using the Kalman gain for the ensemble mean $\mathbf{K}$ as

$$x^{a,det} = x^{b,det} + \mathbf{K}(y^o - H(x^{b,det})) \tag{1}$$

with $x^{a,det}$ and $x^{b,det}$ the deterministic analysis and background, $y^o$ the observation vector,

and $H$ a (non-linear) observation operator (Schraff et al., 2016). KENDA comprises various tools beneficial for ensemble-based DA. Among them are horizontal and vertical observation localization with a Gaspari-Cohn correlation function (Gaspari and Cohn, 1999) using individual length-scales to scale the inverse observation error covariance matrix. Moreover, KENDA allows for analysis calculations on a coarsened grid (Yang et al., 2009) to reduce the

computational costs in the analysis step.



The indirect assimilation of Z observations started at DWD in 2007 with Latent Heat Nudging (LHN; Stephan et al., 2008; Milan et al., 2008), which modifies the thermodynamic model state during model forward integration using low-elevation Z observations. LHN is applicable to both the ensemble and the deterministic run in KENDA. Recently, the direct assimilation of 3D Z

and V observations from the German C-band radar network (see Fig. 1) in combination with LHN became operational in the ICON-D2 routine at DWD.

## 3   Data sets and microphysical retrievals

Intense summer precipitation events can pose a serious risk to society in Central Europe and are particularly difficult to forecast (Olson et al., 1995). Thus, we focus on three intense

summer precipitation events in Germany. The first event covers a 2-day period of heavy, mostly stratiform precipitation over western Germany and its neighboring states from 13 to 14 July 2021, resulting from a slow-moving low-pressure system and causing devastating flooding, e.g., along the Ahr river in North Rhine-Westphalia (case S2021). The second event covers a 3-day period from 24 to 26 July 2017 characterized by widespread intense, mostly

stratiform precipitation. It also caused flooding especially in Lower Saxony in central-northern Germany along the Bode River catchment (case S2017). The third event dominated by convective precipitation covers a 1.5-day period from midday on 19 to 20 July 2017 (case C2017).

### 3.1   Radar observations

DWD operates a network of 16 polarimetric C-band radars (blue circles in Fig. 1) and one additional non-polarimetric radar (red circle). In "volume-scan" mode, the network monitors data consisting of Plan Position Indicators (PPI) at 10 radar elevation angles between 0.5 and 25 degrees with maximum slant ranges of about 180 km every five minutes. The data have a resolution of one kilometer in range, which increased to 0.25 km in March 2021, and one

degree in azimuth; they are taken from the DWD archive.

For the direct assimilation of 3D Z data employed in this study, we use pre-processed Z observations including quality assurance and attenuation correction. For the LWC/IWC estimation, we use the raw polarimetric moments Z (given in dBZ), differential reflectivity ZDR (given in dB), total differential phase PHIDP (given in degrees), and co-polar cross-correlation

coefficient RHOHV. ZDR is the logarithmic ratio between the backscattered power at horizontal and vertical polarizations, which is close to 0 dB for isotropic scatterers and shows larger positive values for oblate particles and negative values for prolate particles. PHIDP is the lag in degrees of the horizontally polarized electromagnetic wave behind the vertically polarized one as the radar signal propagates through the atmosphere filled with anisotropic scatterers

such as raindrops. Typically, half the range-derivative of PHIDP, the specific differential phase shift KDP (given in deg/km), is considered, which is positive for radar volumes filled with oblate



particles and is affected by the presence of liquid water. RHOHV is the cross-correlation coefficient between the horizontally and vertically polarized waves and is thus a measure of the diversity of scatterers in a radar volume. RHOHV decreases in the presence of pronounced

diversity of hydrometeor shapes and in the presence of non-meteorological targets, making it a useful tool for radar data quality assurance.

Kumjian (2013) notes that RHOHV can be as low as 0.85 for snow/ice and 0.95 for rain at S-band. Here, we assume these values also for C-band. Thus, we only consider data below/above the melting layer for RHOHV > 0.95/0.85 with RHOHV corrected for noise before

filtering (Ryzhkov and Zrnic, 2019). The height of the melting layer is determined from so-called Quasi-Vertical Profiles (i.e., azimuthal medians of PPIs measured at sufficiently high elevations and transferred to range-height displays; Trömel et al., 2014; Ryzhkov et al., 2016), as derived from PPIs measured at a 5.5 degree elevation angle, or from the nearest operational DWD radio sounding. KDP is estimated from the filtered and smoothed PHIDP following Vulpiani et

al. (2012) with a fixed window size of nine kilometers. This window size is required due to the rather coarse radial resolution (one kilometer) for most of the PPIs considered to keep noise low and reduce potentially negative KDP estimates. The horizontal specific attenuation A (given in dB km$^{-1}$) – the rate at which power is lost from the transmitted radar signal in horizontal polarization as it propagates through the precipitating atmosphere – is derived below the

melting layer using the filtered and smoothed PHIDP and measured (attenuated) Z using the ZPHI method (Testud et al., 2000). In the retrieval algorithms, the attenuation parameter α (ratio between A and KDP, given in dB deg$^{-1}$) is optimized for each ray using the self-consistency method proposed by Bringi et al. (2001). Finally, the raw Z and ZDR data are corrected for (differential) attenuation using the optimized/climatological α values below/above

the melting layer and the climatological value for the differential attenuation parameter β at C-band 0.02 dB deg$^{-1}$ (Ryzhkov and Zrnic, 2019). For more details on the polarimetric radar moments, see, e.g., Kumjian (2013).

### 3.2   Hybrid liquid water content retrieval

LWC is estimated from the polarimetric radar observations below the melting layer following

the hybrid retrieval proposed by Reimann et al. (2021) developed based on a large pure-rain disdrometer dataset and T-matrix scattering calculations at C-band. The estimator combines different polarimetric radar moments to optimally exploit and mitigate respective advantages and disadvantages known for different precipitation characteristics. For example, in weak precipitation indicated by small total PHIDP increments ΔPHIDP < 5 degrees below the melting

layer, the LWC(Z,ZDR) relation is used (LWC always is in g m$^{-3}$):

$$\log(LWC(Z,ZDR)) = 0.058Z - 0.118ZDR - 2.36. \tag{2a}$$



In such situations, KDP is potentially noisy due to noise in PHIDP and A potentially suffers from an unreliable ΔPHIDP estimation, while the influence of (differential) attenuation on Z and ZDR should be small for these rays. For stronger rain – rays with ΔPHIDP > 5 degrees – the

negative influence of (differential) attenuation on Z and ZDR increases, while less noise and uncertainty is expected in KDP and A; therefore, LWC(A) and LWC(KDP) estimators are used. The LWC(A) estimator

$$\log\big(LWC(A)\big) = -0.1415\log(A)^2 + 0.209\log(A) + 0.46, \tag{2b}$$

is used for radar bins with Z < 45 dBZ, when hail is unlikely, and the LWC(KDP) estimator

$$\log\big(LWC(KDP)\big) = 0.568\log(KDP) + 0.06, \tag{2c}$$

is used for bins with Z > 45 dBZ, since KDP is less affected by hail than A. It should be noted, however, that the hybrid LWC estimator is likely unsuitable in the presence of hail and graupel, especially in certain convective situations, due to its derivation from pure-rain observations.

### 3.3 Hybrid ice water content retrieval

IWC is estimated above the melting using the hybrid estimator proposed by Carlin et al. (2021). It combines the relations based on ZDR and KDP (Ryzhkov and Zrnic, 2019)

$$IWC(zDR, KDP) = 4.0 * 10^{-3} \frac{KDP\lambda}{1 - zDR^{-1}} \tag{3a}$$

with the one based on Z and KDP (Bukovcic et al. 2018, 2020)

$$IWC(z, KDP) = 3.3 * 10^{-2}(KDP\lambda)^{0.67}z^{0.33} \tag{3b}$$

with z and zDR are Z and ZDR given in linear units ($mm^6$ $m^{-3}$ and unitless), IWC in g $m^{-3}$, and the radar wavelength λ set to 53 mm (C-band). The estimators in Eq. (3) are again combined to complement their individual strengths: Eq. (3a) is fairly immune to orientation and shape of snowflakes, but sensitive to variations in ice density and prone to errors from ZDR biases especially at low ZDR values; Eq. (3b) is immune to ZDR miscalibration, but sensitive to

hydrometeor aspect ratio, orientation, and density. Eq. (3a) is used for ZDR > 0.4 dB and Eq. (3b) otherwise. Recently, Blanke et al. (2023) demonstrated the high accuracy of this hybrid estimator (correlation coefficient and root-mean-square deviation 0.96 and 0.19 g $m^{-3}$, respectively) in an evaluation study with in-situ airplane observations on the west coast of the USA. It should be noted, however, that both parts of the hybrid IWC estimator in Eq. (3) are

adapted to snowfall, with their derivation based on an inversely proportional relationship between particle density and diameter, which usually does not hold for hail and graupel. Therefore, its applicability to hail/graupel convective situations in particular may be limited.



## 4  Setup of assimilation experiments

### 4.1  Retrieval resolution

The retrieved LWC and IWC values with the resolution corresponding to the measured radar data are subjected to "superobbing" (see an example in Fig. 2), which is also applied to the Z data in KENDA. Superobbing reduces the resolution of the radar data to approximately match the resolution of the analysis grid by spatial and elevation-wise averaging in the linear scale to a Cartesian grid with a resolution (LC in km) corresponding to the analysis grid (10 km for an

analysis grid coarsening factor of three currently used in KENDA). The number of radar bins contributing to the averaging decreases with increasing distance from the radar, and the window size for the averaging (LS in km) is equal to LC in KENDA, but is modified in our study while keeping LC constant. The minimum number of valid values in the superobbing window to perform superobbing (MV) is three observations, as used for the 3D Z DA in KENDA. Further

details on the superobbing procedure can be found in Bick et al. (2016).

The LWC and IWC estimates are assimilated with a lower limit (LL) similar to the "no-precipitation" threshold of 0 dBZ used for the Z assimilation in KENDA. In contrast to Z, the LWC and IWC data in no-precipitation are mostly filtered out by the applied RHOHV thresholds, but such a lower data threshold can still be useful to limit the variability in the microphysical

estimates and thus can also be used for tuning (personal communication with Ulrich Blahak, DWD). We choose LL = -2.3 for log(LWC), which approximately corresponds to 0 dBZ for Z when comparing measured log(LWC) and synthetic Z data obtained from T-matrix scattering calculations for a large German pure-rain disdrometer data set (not shown). The rare occurrence of snow on the ground in Germany and instrumental limitations prevent a similar

analysis for IWC. Therefore, we also use -2.3 for log(IWC).

Analogous to the assimilation of 3D Z data in KENDA, only the PPIs at radar elevation angles of 1.5, 3.5, 5.5, 8.0, and 12.0 degrees are used for LWC and IWC, and data from altitudes below 600 and above 9,000 m are not considered. The superobbed microphysical estimates are assimilated in the logarithmic scale, similar to the Z data in KENDA, which leads to better

results (not shown).

### 4.2  Assimilation settings and first guess

Z is currently assimilated in KENDA with a fixed observation error standard deviation (OE) of 10 dBZ. We use a fixed value of OE = 0.5, which can be obtained statistically from the disdrometer data considered above: a difference Δlog(LWC) = 0.5 covers a similar fraction of

the full range of data as ΔZ = 10 dBZ (not shown). This value is also used for log(IWC). The horizontal observation localization length-scale (LH) and the vertical observation localization



length-scale (LV) are set to 16 km and to increase with height from 0.075 to 0.5 in logarithm of pressure (ln(p)) as used for the 3D Z DA in KENDA.

First guesses of LWC and IWC are calculated with a simple "forward operator", which uses the
prognostic model variables total air density ($\rho_{tot}$, given in kg m$^{-3}$) and the rain and cloud water mixing ratios $q_r$ and $q_c$ for LWC, and the snow, graupel, and cloud ice mixing ratios $q_s$, $q_g$, and $q_i$ (all given in g m$^{-3}$) for IWC at the model grid points via

$$LWC = 10^3 \rho_{tot}(q_r + q_c) \tag{4a}$$

and

$$IWC = 10^3 \rho_{tot}(q_s + q_g + q_i). \tag{4b}$$

The first-guess LWCs and IWCs are then projected with the nearest-neighbor method onto the polar (PPI) grid of the observed LWC and IWC data and superobbed analogously to the observed data. This procedure is done for the ensemble and the deterministic run.

### 4.3 Model initialization and lateral boundary data

ICON-D2 model data in 40+1-mode for our evaluation periods are provided by DWD for 22 UTC 12 July 2021, 00 UTC 23 July, and 00 UTC 18 July 2017. These data are output from the regular ICON-D2 routine and thus do not require further "spin-up" integrations prior to our assimilation experiments. Hourly assimilation cycles such as in the operational routine including DA of conventional (e.g., surface station, radio sounding, and aircraft data) and 3D
radar observations, and including LHN, are performed to obtain model states for the initial times of the experiment periods 00 UTC 13 July 2021, 00 UTC 24 July 2017, and 11 UTC 19 July 2017. ICON-EU model data provided by DWD are used as lateral boundary conditions.

### 4.4 Experiment part A: assimilation configurations

From the model initial times, 3D LWC and IWC estimates are assimilated in hourly assimilation
cycles instead of 3D Z data, where available, to avoid potential problems arising from assimilating the information from the Z data twice. Thus, Z data is always assimilated within the melting layer and in precipitation-free areas, where the LWC and IWC estimates are not available due to the applied RHOHV thresholds. We exclude the assimilation of 3D V observations and LHN to focus on the assimilation of microphysical information from the radar
network. We assimilate the LWC and IWC estimates separately to study their individual impact on weather forecasts, but also to identify individual best DA parameter (DAP; LH, LV, OE, LS, LL, and MV) sets. The DA configurations assimilating LWC and IWC also assimilate conventional observations and are therefore referred to as CNV+LWC/Z and CNV+IWC/Z. The DA of only conventional observations and the DA of conventional and 3D Z observations are
used as reference configurations CNV and CNV+Z, respectively.





We consider a near-random sample of DAP settings generated via Latin Hypercube Sampling (LHS) by modifying the DAP values from their pre-selected values (pre-selected and modified values in Table 1; generated settings S1-01 to S1-12 in Table 2). The results of using the DAP configurations/values are compared with each other in terms of first-guess deterministic and ensemble QPF quality via a single univariate measure – the joint quality score (JQS)

$$JQS_{c/v} = median_w(\Delta_{\text{CNV+Z}}FSS_{norm}[\text{CNV+X/Z}])$$

$$+median_w(\Delta_{\text{CNV+Z}}BSS_{norm}[\text{CNV+X/Z}]). \tag{5}$$

In Eq. (5), FSS is the deterministic Fraction Skill Score (Roberts and Lean 2008; more details in Appendix B), BSS is the Brier Skill Score (following Wilks 2019; see Appendix C) quantifying the ensemble forecast quality, and both quantities are calculated using DWD's RADOLAN (Radar-Online-Aneichung) product (https://opendata.dwd.de/climate_environment/CDC /grids_germany/hourly/radolan/historical/bin/) as verification data; $\Delta_{\text{CNV+Z}}$ denotes differences with respect to the CNV+Z configuration; X is LWC or IWC; index "norm" denotes normalization with the means of $\Delta_{\text{CNV+Z}}FSS[\text{CNV+Z}]$ or $\Delta_{\text{CNV+Z}}BSS[\text{CNV+Z}]$; $median_w(\dots)$ denotes the weighted median. Medians are used instead of means in order to reduce the impact of outliers in FSS and BSS, and weights are determined by the fractions of threshold exceedances for a given time and threshold of the total number of exceedances at all thresholds (0.5, 1.0, 2.0, and 4.0 mm h$^{-1}$) and events (C2017, S2017, and S2021) in the RADOLAN data (see Fig. 3). We use weighted medians over all cases and thresholds to compare QPF quality between different DAP configurations ($JQS_c$) and additionally calculate weighted medians over all DAP settings that have the same DAP values to compare individual DAP values ($JQS_v$).

In addition to optimizing DAP sets, we also aim to optimally combine the radar data sets considered (i.e., Z, LWC, and IWC). Therefore, also the parallel assimilation of LWC or IWC and Z (configurations CNV+LWC+Z or CNV+IWC+Z, respectively), the combined assimilation of LWC and IWC estimates as alternatives to Z (configuration CNV+[LWC+IWC]/Z) or in parallel to Z (CNV+LWC+IWC+Z) are also evaluated with $JQS_c$.

### 4.5 Experiment part B: nine-hour forecasts

Finally, the impact of assimilating the 3D microphysical estimates with KENDA on forecasts with lead times greater than one hour is evaluated. The 3D LWC and IWC estimates are assimilated with the identified best DAP sets and radar data set configurations in hourly assimilation cycles, as before, and then nine-hour deterministic forecasts of the ICON-D2 model are initiated every third hour from the produced analyses. The quality of the deterministic nine-hour QPF is assessed using the FSS and the Frequency Bias (FBI; more details in Appendix D). Probabilistic forecasts are not considered due to data storage limitations.



## 5 Numerical results

### 5.1 Experiment part A: assimilation configurations

The CNV+LWC/Z configuration yields different first-guess FSS and BSS values for the different DAP settings (see Table 2) and precipitation cases (Fig. 4a, c). Improvements over the assimilation of Z data alone (CNV+Z) considering all cases together are obtained, e.g., with the DAP sets S1-01 to S1-03, or S1-06 (Fig. 4a4, c4). These best-performing sets all have rather small horizontal observation localizations LH of 8 and 16 km and rather high lower limits LL of -1.15 and -2.30 (see Table 2), which may be due to discrepancies between true and model microphysics. Similarly, the IWC assimilation instead of Z where available (CNV+IWC/Z) also yields different first-guess FSS and BSS values for different DAP sets (Table 2) and precipitation cases (Fig. 4b, d). Improvements over the CNV+Z configuration are mostly limited to the 2021 stratiform case, e.g., for the DAP settings S1-02 or S1-05 (Fig. 4b3, d3), while first-guess QPF is mostly degraded for the 2017 convective case (Fig. 4b1, d1).

The univariate measure $JQS_v$ (see Sect. 4.4 and Eq. (5)), which uses the first-guess FSS and BSS values, is used to find the best DAP settings for LWC and IWC. The DAP values LH = 32 km, LV = 0.5 ln(p), OE = 0.5, LS = 5 km, LL = -4.6, and MV = 25 % (i.e., 25 % of the radar pixels in the superobbing window must have valid values) give the worst (and negative) $JQS_v$ values for both LWC and IWC (blue and orange bars in Fig. 5a). Another 10 DAP sets in the vicinity of the better performing ones are sampled with LHS (S2-01 through S2-10 in Table 2). Further improvements over the assimilation of Z alone (CNV+Z) are obtained for the LWC assimilation (Fig. 4e, g), but are mostly only obtained for the 2021 stratiform case for the IWC assimilation (Fig. 4f3, h3). The new DAP settings (Table 2; Fig. 4e-h) do, however, on average not perform significantly better compared to the first sample (Table 2; Fig. 4a-d), except that strong negative outliers (e.g., S1-09 in Fig. 4a-d) do not appear anymore.

The 22 DAP settings (Table 2) for the LWC and IWC assimilations are compared to each other in terms of first-guess deterministic and ensemble QPF quality using the univariate measure $JQS_c$ (see Sect. 4.4 and Eq. (5)). Several DAP settings for the LWC assimilation yield positive $JQS_c$ values (black bars in Fig. 5b) and thus improved first-guess FSS and BSS values compared to the assimilation of Z alone (CNV+Z), while for the IWC assimilation, positive $JQS_c$ values are limited to the 2021 stratiform case (red bars in Fig. 5e). The DAP set S2-06 (LH = 8 km, LV = 0.2 ln(p), OE = 0.25, LS = 20 km, LL = -1.15, and MV = 3, see Table 2) for LWC yields overall the best $JQS_c$ (black bars Fig. 5b), while setting S1-02 (LH = 8 km, LV = 0.5 ln(p), OE = 0.25, LS = 10 km, LL = -1.15, and MV = 50 %, see Table 2) results in the best (but rather neutral) $JQS_c$ value for IWC (red bars in Fig. 5b).



The assimilation of LWC instead of Z data where possible (CNV+LWC/Z) with the respective

best DAP setting improves first guess QPF for the 2017 precipitation cases (Fig. 4e1, e2, g1, g2 and black bars in Fig. 5c, d) compared to the assimilation of Z data alone (CNV+Z) while QPF quality is degraded for the stratiform S2021 case (Fig. 4e3, g3 and black bars in Fig. 5e). As expected, the time series of the first-guess FSS and BSS values at a threshold of 0.5 mm h$^{-1}$ show slight, systematic improvements for the 2017 cases for some time intervals (green colors

in Fig. 6a, c, e, g), but more pronounced degradations for the 2021 case (Fig. 6i, k). The assimilation of IWC (CNV+IWC/Z) with the respective best DAP set yields improvements over the CNV+Z configuration particularly for the stratiform S2021 case (Fig. 4b3, d3 and red bars in Fig. 5e), but clear quality decreases for the convective C2017 case (Fig. 4b1, d1 and red bars in Fig. 5c). Time series of first-guess FSS and BSS values at a 0.5 mm h$^{-1}$ threshold

confirm this finding: slight, systematic improvements are evident for the 2021 case in some time periods (Fig. 6j, l), while degradations are visible for the 2017 convective case (Fig. 6b, d). The better performance of the IWC assimilation for the 2021 stratiform case may be due the higher radial resolution of the more recent radar data of DWD (recall that the resolution was increased from one kilometer to 0.25 km in spring 2021), which leads to better KDP estimates,

because many more consecutive radar bins are considered for the nine-kilometer KDP-estimation window used. Using the same window length for the lower-resolution data for the 2017 cases means using only one quarter of the data compared to the 2021 case. Estimating KDP from only nine consecutive values may favor negative KDP estimates resulting in negative IWC values, which are set to the lower limit (LL) value in the superobbing procedure

and are thus treated as "no-precipitation". The replacement of negative IWC estimates with zero or with the IWC(Z) retrievals following Atlas et al. (1995) led to some improvements, but the first-guess QPF quality remained below the CNV+Z configuration (not shown).

Parallel assimilation of LWC and Z (CNV+LWC+Z), i.e., assimilation of LWC and Z at the same superobbing points, reduces the JQS$_c$ values compared to the alternative assimilation strategy

(CNV+LWC/Z), but is still better than the assimilation of Z only (CNV+Z; lower black bars in Fig. 7). In contrast, the parallel assimilation of IWC and Z (CNV+IWC+Z) improves JQS$_c$ values compared to the alternative assimilation strategy (CNV+IWC/Z; middle black bars in Fig. 7) above the CNV+Z quality. Assimilation of all radar data sets in parallel (CNV+LWC+IWC+Z) gives the best JQS$_c$ value (upper black bar in Fig. 7b).

The impact of the LWC and IWC assimilation on the first-guess of temperature, relative humidity, and u-wind speed is investigated using conventional observations. The assimilation of radar information generally reduces standard deviations (SD) compared to the assimilation of only conventional data (CNV+Z, CNV+LWC/Z, CNV+IWC/Z, and CNV+LWC+IWC+Z configurations correspond to black, red, yellow, and blue curves in Fig. 8b, e, h), while the

impact on mean bias deviations (MBD) is less clear (compare black solid, red, yellow, and blue





curves with black dotted curves in Fig. 8c, f, i). The CNV+LWC/Z, CNV+IWC/Z, and CNV+LWC+IWC+Z configurations result in SDs and MBDs similar to the assimilation of Z alone (CNV+Z), but slight, systematic SD improvements are evident for the u-wind speed with the CNV+IWC/Z configuration (yellow curve in Fig. 8h).

### 5.2 Experiment part B: nine-hour forecasts

With the best performing DAP sets for the LWC and IWC assimilations, up to nine-hour forecasts are performed. Z observations (CNV+Z) clearly improve the deterministic FSS for a threshold of 0.5 mm h$^{-1}$ for all forecast hours compared to the assimilation of only conventional data (CNV) on average for all cases (compare black with grey lines in Fig. 9a, d, g, j). This also holds for the deterministic FBI for the stratiform S2017 and S2021 cases, while for the convective C2017 case the underestimation is enhanced (compare black and grey curves in Fig. 9c, f, i, l). Assimilating LWC estimates instead of Z data where possible (CNV+LWC/Z) slightly further improves the FSS on average over all cases for most of the forecast time (red curve above the zero line in Fig. 9b). This overall positive impact results from the first six hours of the convective C2017 case and forecast hours five to nine of the stratiform 2021 case (Fig. 9e, k). FBI improvements are achieved for up to seven hours lead time (compare red with black curves in Fig. 9c) and at least for the first four hours lead time for all individual cases (compare red curves with grey and black curves in Fig. 9f, i, l).

The assimilation of IWC instead of Z where possible (CNV+IWC/Z) only marginally improves the FSS on average for the first five hours lead time (yellow curves in Fig. 9b) compared to the CNV+Z configuration. As expected from the first-guess analysis, the mean FSS for the convective C2017 case is mostly degraded (yellow curve in Fig. 9e) and the stratiform S2017 and S2021 cases are improved (yellow curves in Fig. 9h, k). For the S2021 case, the mean forecast FSS values are slightly improved for most of the forecast time (yellow curve mostly above zero line in Fig. 9k). Qualitatively similar results result for the FBI on average over all cases, which shows the best results for the first four forecast hours (compare yellow with the remaining curves in Fig. 9c).

The on average best FSS for the first six forecast hours are obtained, when all radar data sets are assimilated together (CNV+LWC+IWC+Z; blue curve in Fig. 9b); however, the good results for the FBI with the assimilation of IWC (CNV+IWC/Z) are not reached (compare blue and yellow curves in Fig. 9c), but the FBI is improved up to seven forecast hours compared to the CNV+Z configuration (black curve).

As expected, the SDs of 2m temperature, 2m relative humidity, and 10m u-wind speed generally increase with forecast lead time for all DA configurations (CNV, CNV+Z, CNV+LWC/Z, CNV+IWC/Z, and CNV+LWC+IWC+Z drawn as grey, black, red, yellow, and blue curves, respectively, in Fig. 10). The assimilation of radar information always reduces the





SDs. Interestingly, the assimilation of IWC yields the lowest SD for humidity (yellow curve in Fig. 10c) and wind (Fig. 10e) and is only marginally outperformed by the assimilation of all radar information in parallel (CNV+LWC+IWC+Z) for 2m temperature (compared yellow with blue curve in Fig. 10a). The bias (MBD), however, is only reduced for the near-surface wind (Fig. 10f), while the absolute MBD generally increases due to the assimilation of radar data – except for the near-surface humidity, which achieves its lowest values when all radar information is assimilated in parallel (CNV+LWC+IWC+Z; blue curve in Fig. 10d).

## 6 Conclusions

We assimilated for the first time polarimetric information from radar observations of the German C-band radar network in the KENDA-ICON-D2 system of DWD. In this study, we used microphysical retrievals of liquid and ice water content (LWC and IWC) and evaluated their impact on short-term precipitation forecasts. First, the impact of assimilating the microphysical retrievals on the first-guess (hourly) precipitation forecasts was investigated with different data assimilation parameter (DAP; e.g., observation localization length-scales and errors) sets and radar data set configurations. Then, the most successful assimilation settings were used to initiate nine-hour precipitation forecasts.

Four data set configurations were analyzed for finding the best DAP sets: only conventional observations (CNV), conventional and 3D reflectivity Z observations (CNV+Z), conventional data and 3D LWC estimates replacing Z observations where available (CNV+LWC/Z), and conventional data and 3D IWC estimates replacing Z observations where possible (CNV+IWC/Z). For the two stratiform cases in the summers of 2017 and 2021 and the one convective case in the summer of 2017, a rather small horizontal observation localization length-scale of 8 km and a lower limit of -1.15 in log(LWC) and log(IWC) yielded the best deterministic and ensemble first-guesses. Thus, best precipitation forecasts are achieved when the influence of the observed microphysical estimates on the model state is rather small, possibly due to discrepancies between model and true microphysics. A rather small observation error standard deviation of 0.25 in log(LWC) and log(IWC) was most successful. The best values for the other DAPs differed for LWC and IWC: vertical localization length-scales were 0.2 in logarithm of pressure for LWC and 0.5 in logarithm of pressure for IWC; best superobbing window sizes were 20 km for LWC and 10 km for IWC; the minimum number of valid values in the superobbing window was three observations for LWC and 50 % valid values for IWC.

The LWC assimilation with the best performing DAP setting improved the first-guesses for most precipitation cases and accumulation thresholds compared to the assimilation of Z alone (CNV+Z), while the best-performing DAP setting for IWC deteriorated the results, especially for the 2017 convective case, except for the stratiform case in 2021. The latter may be due to



the radial resolution increase in the DWD volume scans from one kilometer to 0.25 km in spring
2021. The higher resolution improves the specific differential phase KDP estimation as part of

525     the hybrid IWC retrieval, because more successive radar bins can be used for a given KDP
window size. One reason for the poor performance of the IWC assimilation especially for the
2017 convective case, besides possible deficiencies in the model's ice module, may be the
fact that the IWC retrieval was adjusted for snowfall but not for hail or graupel likely being
present during intense convective summer precipitation in Germany. Interestingly, the LWC

assimilation led to consistent improvements for convective situations, despite a retrieval not
adapted to hail or graupel either. The application of a higher co-polar cross-correlation
coefficient RHOHV threshold below the melting layer for filtering may have masked radar pixels
contaminated with hail or graupel.

In general, the best first-guess precipitation forecasts were obtained when all radar data sets

(i.e., Z, LWC, and IWC) were assimilated together (CNV+LWC+IWC+Z).

Nine-hour forecasts initiated with the CNV+LWC/Z configuration using the best DAP setting
slightly outperformed the assimilation of Z data alone (CNV+Z) in terms of deterministic
Fraction Skill Score FSS on average and for most forecast lead times with the best results for
the 2017 convective case. The same applies to the assimilation of IWC (CNV+IWC/Z),

however, the mean FSS mostly deteriorated for the convective case compared to the CNV+Z
configuration, but was systematically improved over most of the forecast time for the high-
resolution 2021 stratiform case. Forecasts initiated with the assimilation of all radar data sets
(CNV+LWC+IWC+Z) yielded the best overall FSS. Furthermore, the assimilation of the LWC
and/or IWC estimates (CNV+LWC/Z, CNV+IWC/Z, and CNV+LWC+IWC+Z) generally

improved the mean frequency bias FBI over the assimilation of Z alone (CNV+Z) for most
forecast hours.

We used DWD's standard configuration of KENDA, which only produces microphysical
analysis increments in cloud water mixing ratio and specific humidity, i.e., not all available
hydrometeor species (e.g., rain, cloud ice, and graupel mixing ratios) are updated individually.

This setting was chosen at DWD to optimize the assimilation impact of Z (personal
communication with Klaus Stephan, DWD). Thus, it remains to be explored how changes in
the updated (microphysical) variables change precipitation forecasts when polarimetric
information contained in microphysical retrievals is assimilated. For example, it should be
investigated if the update of the rain mixing ratio via cross-correlations in the first-guess

ensemble from LWC observation increments or the update of ice species (e.g., snow and/or
cloud-ice mixing ratios) via cross-correlations from IWC innovations would yield improved
forecasts.





Our study presented the benefits from the assimilation of state-of-the-art polarimetric microphysical retrievals below and above the melting layer adjusted for pure rain and snowfall, respectively, in a convective-scale NWP system in Germany. The results revealed only limited benefits with the assimilation of IWC retrievals in convective precipitation. Since the retrievals are based on assumptions valid for snow but not for graupel or hail, such as e.g. the inversely proportional relationship between density and size of hydrometeors, the potential presence of graupel and/or hail in convection may be at least partly responsible. Accordingly, the development of more adequate retrieval algorithms for convective cores constitutes one of the next steps to further improve the exploitation of ice microphysical retrievals for radar data assimilation.

**Appendices**

**Appendix A: Local Ensemble Transform Kalman Filter**

The Local Ensemble Transform Kalman Filter (LETKF; Hunt et al., 2007) uses an ensemble of background model states each of dimension N

$$\{ \boldsymbol{x}_k^{b,m} : m = 1, 2, \dots, M \} \tag{A1}$$

at time $t_k$, with M the ensemble size, resulting from the forward integration of an ensemble of analyses

$$\{ \boldsymbol{x}_{k-1}^{a,m} : m = 1, 2, \dots, M \} \tag{A2}$$

at time $t_{k-1}$. In the following formulations we refer to time step $t_k$ and drop the time index for simplicity. The mean and covariance matrix associated with the background ensemble are given by

$$\overline{\boldsymbol{x}}^b = M^{-1} \sum_{m=0}^{M} \boldsymbol{x}^{b,m} \tag{A3}$$

and

$$\mathbf{P}^b = (M-1)^{-1} \mathbf{X}^b (\mathbf{X}^b)^T, \tag{A4}$$

with $\mathbf{X}^b$ a $N \times M$-matrix the columns of which are the perturbations of the individual background ensemble members from the respective background ensemble mean as

$$\mathbf{X}^b = \begin{bmatrix} x^{b,n=0,m=0,} - \overline{x}^{b,n=0} & \cdots & x^{b,n=0,m=M} - \overline{x}^{b,n=0} \\ \vdots & \ddots & \vdots \\ x^{b,n=N,m=0} - \overline{x}^{b,n=N} & \cdots & x^{b,n=N,m=M} - \overline{x}^{b,n=N} \end{bmatrix}. \tag{A5}$$

In the LETKF analysis, an ensemble of analyses such as in Eq. (A2) is constructed at time $t_k$ such that the associated ensemble mean and covariance matrix are given by





$$\overline{\boldsymbol{x}}^a = M^{-1} \sum_{m=0}^{M} \boldsymbol{x}^{a,m} \tag{A6}$$

and

$$\mathbf{P}^a = (M-1)^{-1} \mathbf{X}^a (\mathbf{X}^a)^T, \tag{A7}$$

with the columns of the $N \times M$-matrix $\mathbf{X}^a$, like $\mathbf{X}^b$, the perturbations of the individual analysis ensemble members from their respective analysis ensemble mean. The analysis increment is determined in the M-dimensional subspace spanned by the background ensemble perturbations or columns of $\mathbf{X}^b$ by minimizing the cost function

$$\tilde{J}(\boldsymbol{w}) = (M-1)\boldsymbol{w}^T\boldsymbol{w} + \left(\boldsymbol{y}^o - H(\overline{\boldsymbol{x}}^b + \mathbf{X}^b\boldsymbol{w})\right)^T \mathbf{R}^{-1}(\boldsymbol{y}^o - H(\overline{\boldsymbol{x}}^b + \mathbf{X}^b\boldsymbol{w})). \tag{A8}$$

Here, the vector $\boldsymbol{w} \in R^M$ determines a model state $\boldsymbol{x}$ through a linear combination of the background ensemble perturbations via

$$\boldsymbol{x} = \overline{\boldsymbol{x}}^b + \mathbf{X}^b\boldsymbol{w}. \tag{A9}$$

$\boldsymbol{y}^o$ in Eq. (A8) denotes the P-dimensional observation vector, the $P \times P$-matrix $\mathbf{R}$ is the corresponding covariance matrix, and $H$ is the observation operator. In the LETKF, $H$ is

linearized about the background ensemble mean as

$$H(\overline{\boldsymbol{x}}^b + \boldsymbol{X}^b\boldsymbol{w}) \approx \overline{\boldsymbol{y}}^b + \mathbf{Y}^b\boldsymbol{w} \tag{A10}$$

with $\overline{\boldsymbol{y}}^b$ the ensemble mean of the background ensemble in observation space and $\mathbf{Y}^b$ the corresponding $P \times M$-matrix of observation-background ensemble perturbations. Applying the linearization in the cost function formulation in Eq. (A8) yields

$$\tilde{J}^*(\boldsymbol{w}) = (M-1)\boldsymbol{w}^T\boldsymbol{w} + \left(\boldsymbol{y}^o - \overline{\boldsymbol{y}}^b + \mathbf{Y}^b\boldsymbol{w}\right)^T \mathbf{R}^{-1}(\boldsymbol{y}^o - \overline{\boldsymbol{y}}^b + \mathbf{Y}^b\boldsymbol{w}), \tag{A11}$$

and the minimum of $\tilde{J}^*$ can be explicitly calculated due to its formulation in the low-dimensional ensemble space. We yield the mean and covariance matrix in ensemble space

$$\overline{\boldsymbol{w}}^a = \widetilde{\mathbf{P}}^a (\mathbf{Y}^b)^T \mathbf{R}^{-1}(\boldsymbol{y}^o - \overline{\boldsymbol{y}}^b) \tag{A12}$$

and

$$\widetilde{\mathbf{P}}^a = ((M-1)\mathbf{I} + (\mathbf{Y}^b)^T \mathbf{R}^{-1} \mathbf{Y}^b)^{-1}, \tag{A13}$$

and the corresponding mean and covariance matrix in the full N-dimensional model space

$$\overline{\boldsymbol{x}}^a = \overline{\boldsymbol{x}}^b + \mathbf{X}^b \overline{\boldsymbol{w}}^a = \overline{\boldsymbol{x}}^b + \mathbf{X}^b \widetilde{\mathbf{P}}^a (\mathbf{Y}^b)^T \mathbf{R}^{-1}(\boldsymbol{y}^o - \overline{\boldsymbol{y}}^b) \tag{A14}$$

and

$$\mathbf{P}^a = \mathbf{X}^b \widetilde{\mathbf{P}}^a (\mathbf{X}^b)^T. \tag{A15}$$





Thus, the analysis ensemble mean $\bar{x}^a$ is calculated by adding to the background ensemble mean $\bar{\mathbf{x}}^b$ the innovation or observation increment $y^o - \bar{y}^b$ weighted by the Kalman gain $\mathbf{K} = \mathbf{X}^b \widetilde{\mathbf{P}}^a (\mathbf{Y}^b)^T \mathbf{R}^{-1}$. The individual analysis ensemble members are determined using a symmetric square root

$$\mathbf{X}^a = \mathbf{X}^b \mathbf{W}^a \tag{A16}$$

with

$$\mathbf{W}^a = ((M-1)\widetilde{\mathbf{P}}^a)^{1/2} \tag{A17}$$

such that

$$x^{a,m} = \bar{x}^b + \mathbf{X}^b(\bar{w}^a + \mathbf{W}_m^a) \tag{A18}$$

with $W_m^a$ the m-th column of $\mathbf{W}^a$.

**Appendix B: Fraction Skill Score (FSS)**

The Fraction Skill Score (FSS; Roberts and Lean, 2008) is calculated via projection of the forecasted and observed precipitation accumulations onto the verification grid (the RADOLAN grid reduced to three kilometers to better fit the model data of about 2.2 km horizontal resolution). The RADOLAN data are averaged over nine grid points, while the model data are

selected by the nearest-neighbor method. The projected fields of observations $P_O$ and model first-guess $P_M$ are converted to binary fields $I_O$ and $I_M$ for the chosen precipitation accumulation thresholds $q$

$$I_{O,(q)} = \begin{cases} 1 & for \ P_O \geq q \\ 0 & for \ P_O < q \end{cases} \tag{B1}$$

and

$$I_{M,(q)} = \begin{cases} 1 & for \ P_M \geq q \\ 0 & for \ P_M < q \end{cases}. \tag{B2}$$

Fractions of surrounding points within squares of $n \times n$ data points in the binary fields $I_{O,(q)}$ and $I_{M,(q)}$, $F_{O,(n,q)}$ and $F_{M,(n,q)}$, that have a value of one are calculated for each verification grid point. Finally, the FSS for a window size n and precipitation threshold q is computed as

$$FSS_{(n,q)} = 1 - \frac{MSD_{(n,q)}}{MSD_{(n,q),ref}} \tag{B3}$$

with the mean squared deviation (MSD) for the observation and forecast fractions



$$MSD_{(n,q)} = \frac{1}{N} \sum_{i=1}^{N} \left[ F_{O,(n,q),i} - F_{M,(n,q),i} \right]^2 \tag{B4}$$

and the total number of verification grid points N. The reference MSD

$$MSD_{(n,q),ref} = \frac{1}{N} \sum_{i=1}^{N} F_{O,(n,q)i}^2 + F_{M,(n,q)i}^2 \tag{B5}$$

represents the largest possible MSD from the observation and forecast fractions. The FSS
shows values between zero and one with the higher values the better. In this paper, n is chosen
to be five corresponding to a 15 km window.

**Appendix C: Brier Skill Score (BSS)**

The Brier Score (BS; Wilks, 2019) is a measure for the accuracy of probabilistic forecasts and
takes the forecast ensemble into account via

$$BS_{(q)} = \frac{1}{N} \sum_{i=1}^{N} \left[ p_{(q),i} - I_{O,(q),i} \right]^2 \tag{C1}$$

with $p_{(q),i}$ the fraction of ensemble members within the ensemble exceeding the threshold $q$ at
the $i^{th}$ verification grid point. The Brier Skill Score BSS for a threshold $q$ is then calculated as

$$BSS_{(q)} = 1 - \frac{BS}{BS_{ref}} \tag{C2}$$

with $BS_{ref}$ the Brier score of a reference ensemble forecast (here forecasts resulting from
configuration CNV). The BSS shows positive values if the probabilistic QPF fits the
observations better than the reference QPF and vice versa.

**Appendix D: Frequency Bias (FBI)**

The Frequency Bias (FBI)

$$FBI_{(q)} = \frac{a_{(q)} + b_{(q)}}{a_{(q)} + c_{(q)}} \tag{D1}$$

with $a_{(q)}$ the total number of verification grid points that exceed threshold $q$ in $P_O$ and $P_M$, $b_{(q)}$
the total number of points where $q$ is exceeded in $P_M$ but not in $P_O$, and $c_{(q)}$ the total number
of points for which $q$ is not exceeded in $P_M$ but in $P_O$. The FBI shows values below/above one
in the case of under/overforecasted number of threshold exceedances.




**Code availability**

The experiments were performed at the DWD servers with the ICON-D2 and KENDA codes accessable through the BACY (Basic Cycling) environment at DWD.

**Data availability**

The model and observational data used for the assimilation experiments in this study were provided by the DWD data archive upon request.

**Author contribution**

Lucas Reimann was responsible for writing the original draft, performing the assimilation experiments at the DWD servers, analysing the results, and creating the visualizations.
Clemens Simmer and Silke Trömel directed the study, helped in the analysis process, and reviewed the paper.

**Competing interests**

The authors declare that they have no conflict of interest.

**Special issue**

Fusion of radar polarimetry and numerical atmospheric modelling towards an improved understanding of cloud and precipitation processes

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



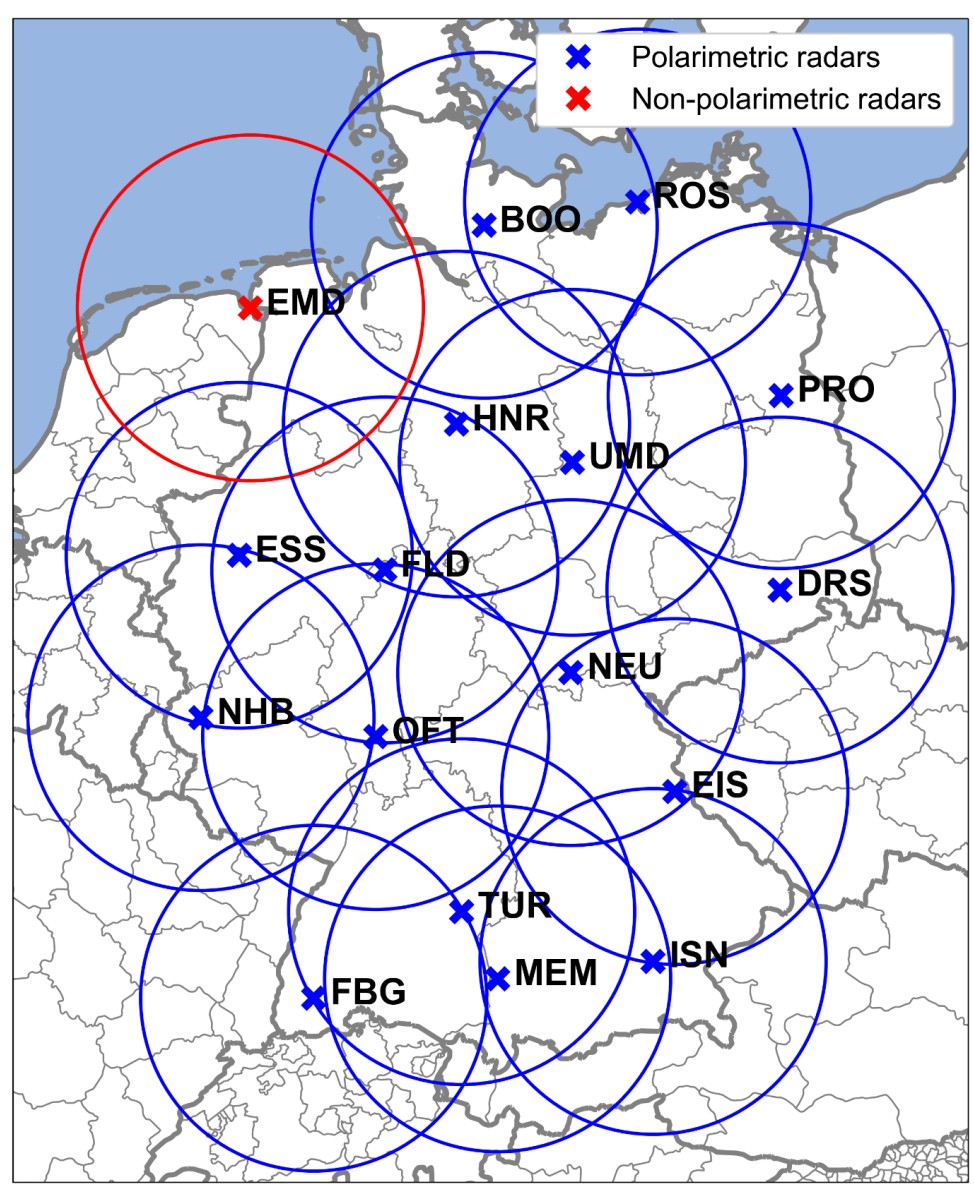

**Figure 1**: German polarimetric C-band radar network operated by DWD. Crosses indicate locations of radar stations in Emden (EMD), Boostedt (BOO), Rostock (ROS), Hannover (HNR), Ummendorf (UMD), Prötzel (PRO), Essen (ESS), Flechtdorf (FLD), Dresden (DRS), Neuhaus (NEU), Neuheilenbach (NHB), Offenthal (OFT), Eisberg (EIS), Türkheim (TUR), Isen (ISN), Memmingen (MEM), and Feldberg (FBG), circles indicate approximate ranges of 150 km around radars; blue color indicates polarimetric and red color indicates non-polarimetric radars.




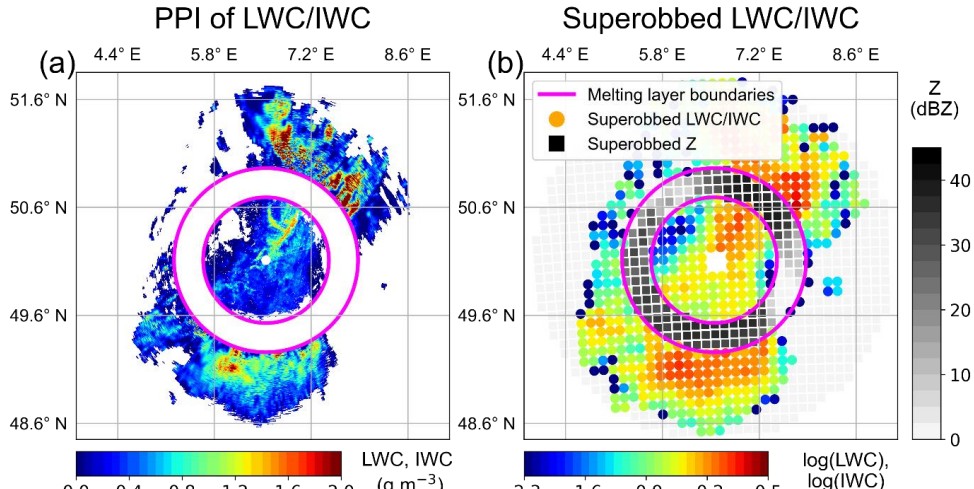

**Figure 2**: Visualization of the superobbing process from (a) a PPI of estimated LWC (Eq. (2)) below and IWC (Eq. (3)) above the melting layer (approximate upper and lower boundaries of the melting layer indicated by violet rings) at 1.5 degrees of the DWD radar NHB (see Fig. 1) for the stratiform precipitation case S2021 at 14 July 2021 16 UTC to (b) the corresponding field of superobbed (with the pre-selected settings LS = 10 km, LL = -2.3, and MV = 3) log(LWC) and log(IWC) (colored dots) and superobbed reflectivity Z (grey squares), where no LWC/IWC estimates are available (e.g., within the melting layer).

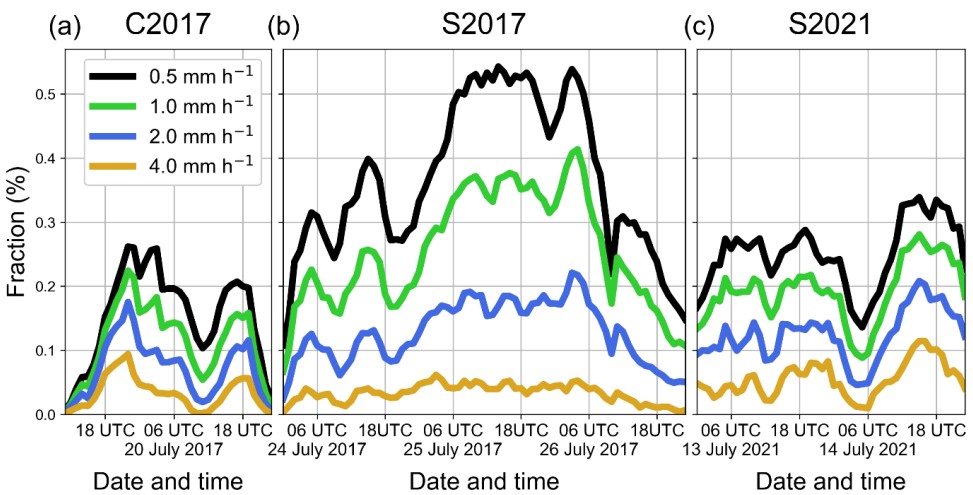

**Figure 3**: Exceedances of hourly rain accumulation thresholds 0.5 (black curves), 1.0 (green), 2.0 (blue), and 4.0 mm h$^{-1}$ (yellow) in the RADOLAN data (hourly accumulations) for the rainfall cases (a) C2017, (b) S2017, and (c) S2021 as percentages of the total number of threshold exceedances in all three rainfall cases and thresholds considered. The fractions are used to determine weights for calculations of weighted medians of FSS and BSS (e.g., in Fig. 4), and for the calculation of the univariate measure JQS (see Eq. (5) in Sect. 4.4).





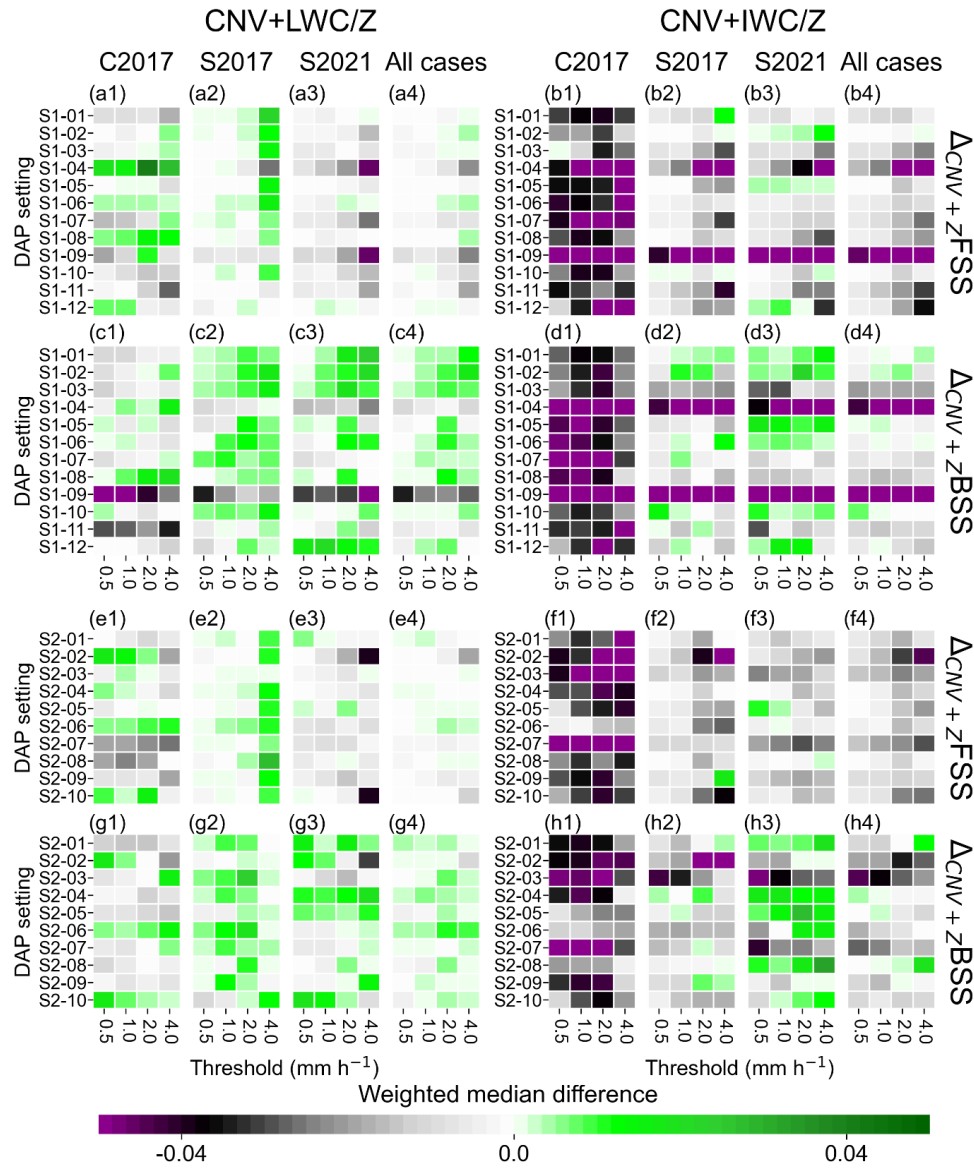

**Figure 4**: Weighted medians of differences in first-guess deterministic FSS (first and third panel rows) and BSS (second and fourth panel rows) between the CNV+LWC/Z (left block) or CNV+IWC/Z (right block) configurations with different sampled DAP settings (S1-01 to S1-12 and S2-01 to S2-10 in Table 2) and the CNV+Z configuration for accumulation thresholds 0.5, 1.0, 2.0, and 4.0 mm h$^{-1}$ and the three rainfall periods considered (three left columns within each block). The right most column in each block shows the weighted median over all cases considered. Weights are determined by threshold exceedances in the RADOLAN data (see Fig. 3). Green color indicates improvements compared to the CNV+Z configuration, grey to dark purple color indicates degradations.



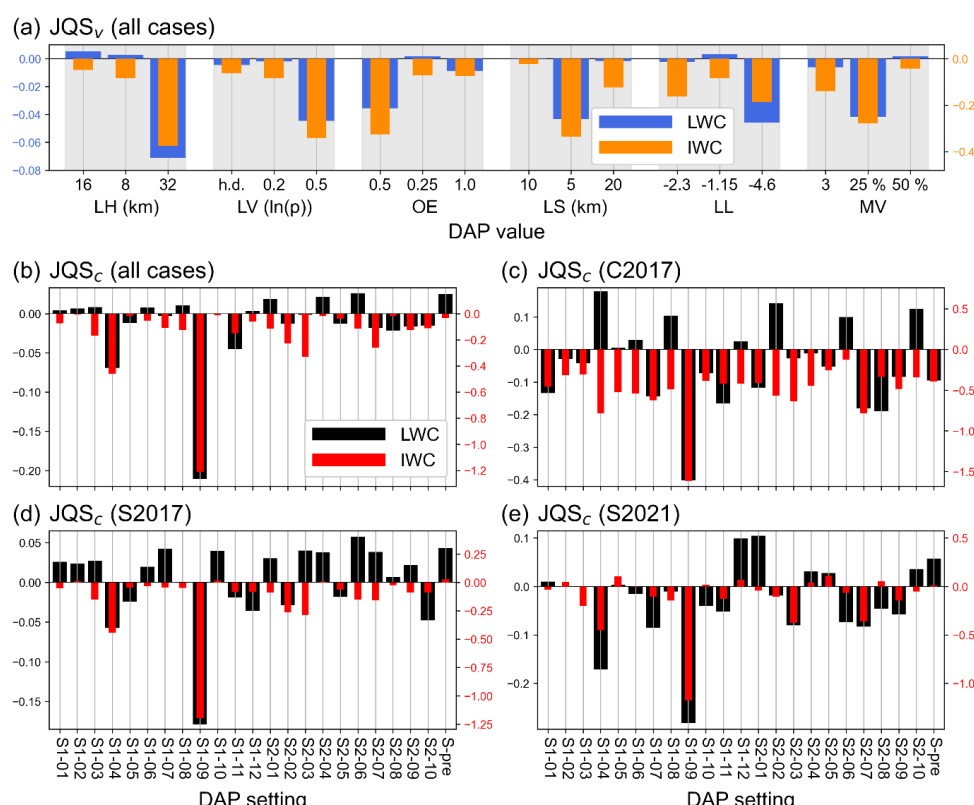

**Figure 5**: (a) Comparison of the investigated DAP values for LH, LV, OE, LS, LL, and MV (Table 1) in terms of the univariate measure $JQS_v$ (see Eq. (5) in Sect. 4.4) for the LWC (blue bars) and IWC (orange bars) assimilation with the DAP settings from the first DAP settings (S1-01 to S1-12 in Table 2). In (b), all 22 DAP settings (S1-01 to S1-12 and S2-01 to S2-10 in Table 2) plus the pre-selected DAP setting (setting S-pre in Table 1) are compared with each other in terms of the univariate measure $JQS_c$ (see Eq. (5) in Sect. 4.4) for the LWC (black bars) and IWC (red bars) assimilation considering all rainfall cases together. Panels (c), (d), and (e) are like panel (b), but with the $JQS_c$ calculated for the individual rainfall cases C2017, S2017, and S2021, respectively.



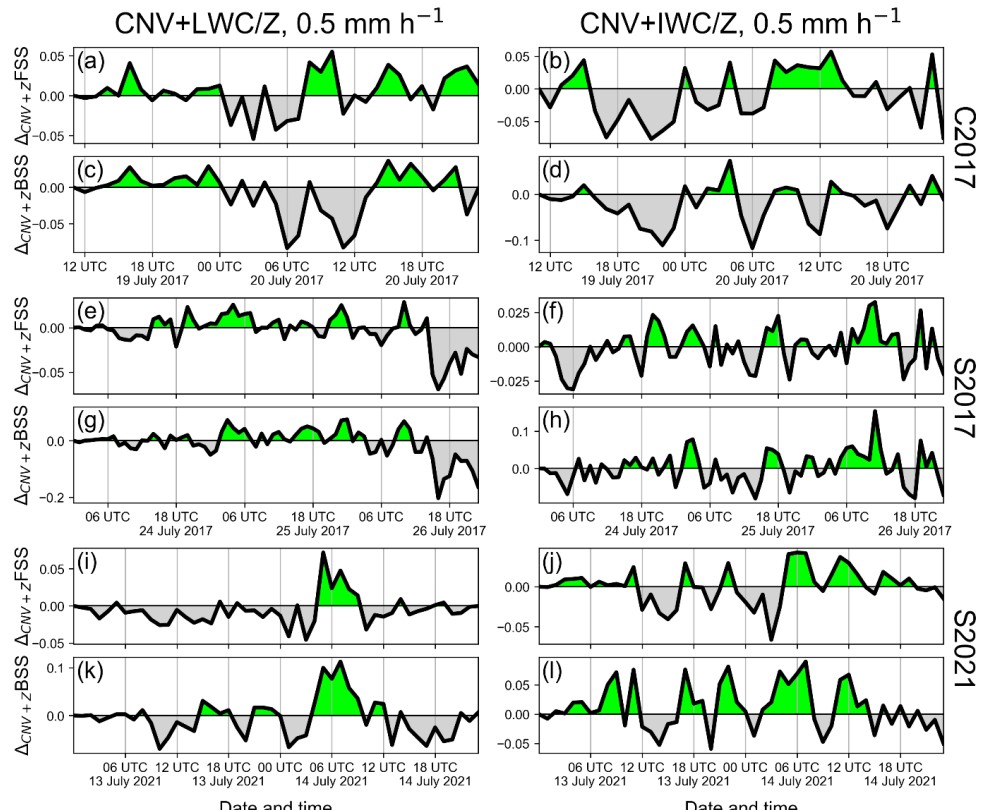

**Figure 6**: Time series of the difference in first-guess deterministic FSS and BSS for a threshold of 0.5 mm h$^{-1}$ between the CNV+LWC/Z (left panel column) or CNV+IWC/Z (right panel column) configurations and the CNV+Z configuration using the found best-performing DAP settings for LWC and IWC (S2-06 and S1-02, see Table 2) for the precipitation cases (a)-(d) C2017, (e)-(h) S2017, and (i)-(l) S2021. Green shading indicates improvements with respect to CNV+Z, grey shading indicates deteriorations.



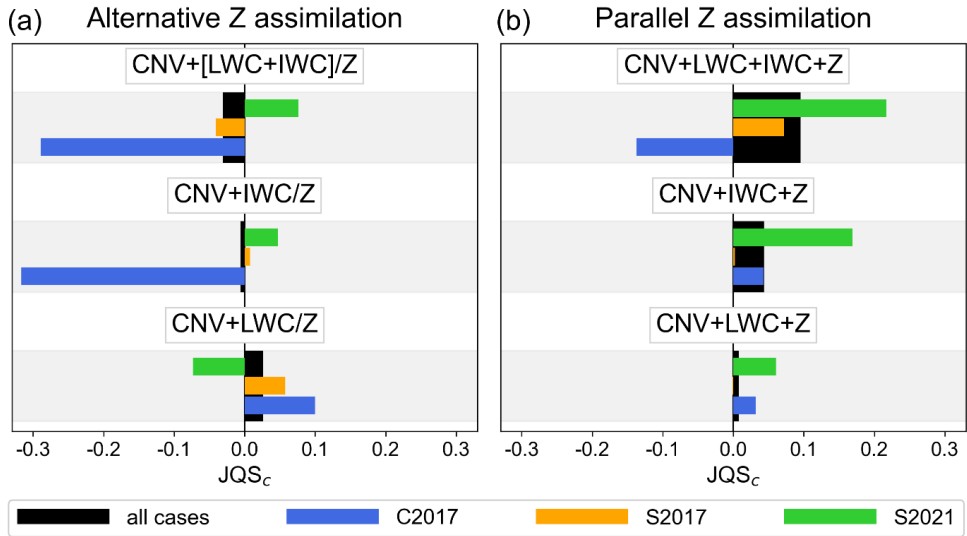

**Figure 7**: Comparison of different radar data set configurations in terms of the univariate measure $JQS_c$ (see Eq. (5) in Sect. 4.4). Configurations assimilating LWC and/or IWC with the found best DAP settings (S2-06 and S1-02 in Table 2) (a) instead of Z where possible (alternative Z assimilation) in configurations CNV+LWC/Z, CNV+IWC/Z, and CNV+[LWC+IWC]/Z (lower, middle, and upper bars), and (b) together with Z (parallel Z assimilation) in configurations CNV+LWC+Z, CNV+IWC+Z, and CNV+LWC+IWC+Z (lower, middle, and upper bars) are compared. Black bars indicate the $JQS_c$ calculated over all three rainfall cases, and blue, orange, and green bars indicate the $JQS_c$ calculated over the individual cases C2017, S2017, and S2021, respectively.





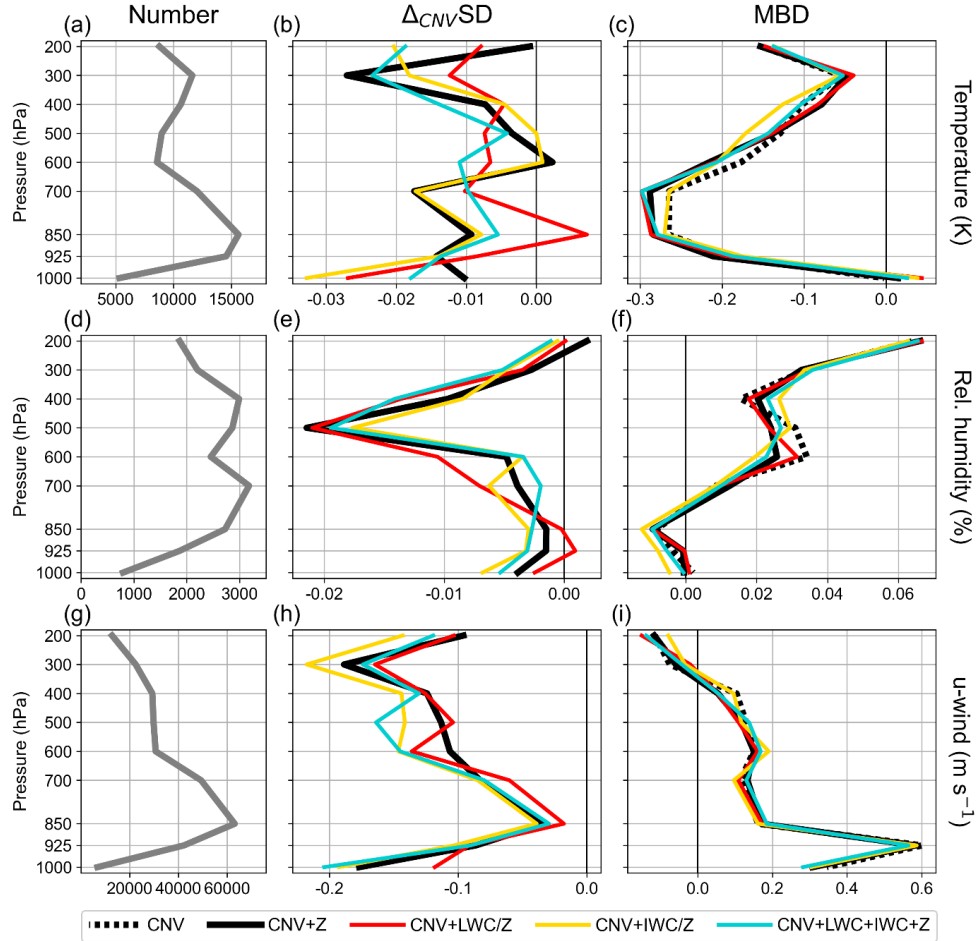

**Figure 8**: Vertical profiles of differences in standard deviations (SD) with respect to the CNV configuration (middle column) and of mean bias deviations (MBD; right column) of first-guesses of temperature (upper row), relative humidity (middle row), and u-wind (lower row) obtained from hourly assimilation cycles with the assimilation configurations CNV (black dotted), CNV+Z (black solid), CNV+LWC/Z (red), CNV+IWC/Z (yellow), and CNV+LWC+IWC+Z (blue curves) from conventional observations over Germany. The number of observations contributing to the SD and MBD calculations are shown in the left column (grey solid curves). All rainfall cases are considered and the found best DAP settings for LWC and IWC (S2-06 and S1-02 in Table 2) are used for the LWC and IWC assimilations.



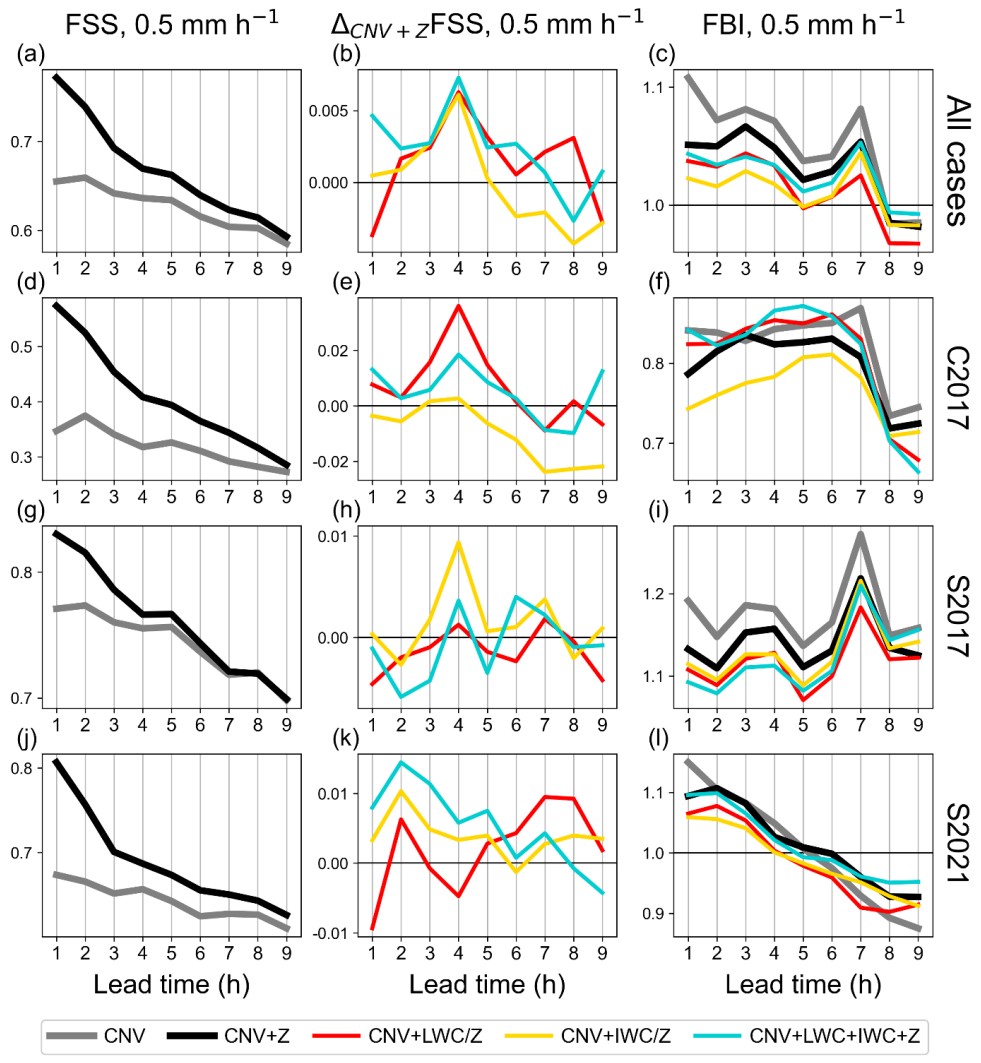

**Figure 9**: Left panel column: time series of the deterministic FSS for a 0.5 mm h[-1] threshold of nine-hour forecasts initiated every third hour from hourly assimilation cycles with the CNV and CNV+Z configurations (grey and black curves) as means over all precipitation cases (upper row), over only the 2017 convective case C2017 (second row), over only 2017 stratiform case S2017 (third row), and over only the 2021 stratiform case S2021 (lower row). Middle column: corresponding deviations in mean deterministic FSS from the CNV+Z configuration of the CNV+LWC/Z (red curves), CNV+IWC/Z (yellow curves), and CNV+LWC+IWC+Z (blue curves) configurations using the found best DAP settings for LWC and IWC assimilations (S2-06 and S1-01 in Table 2). Right column: corresponding mean deterministic FBI.

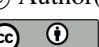



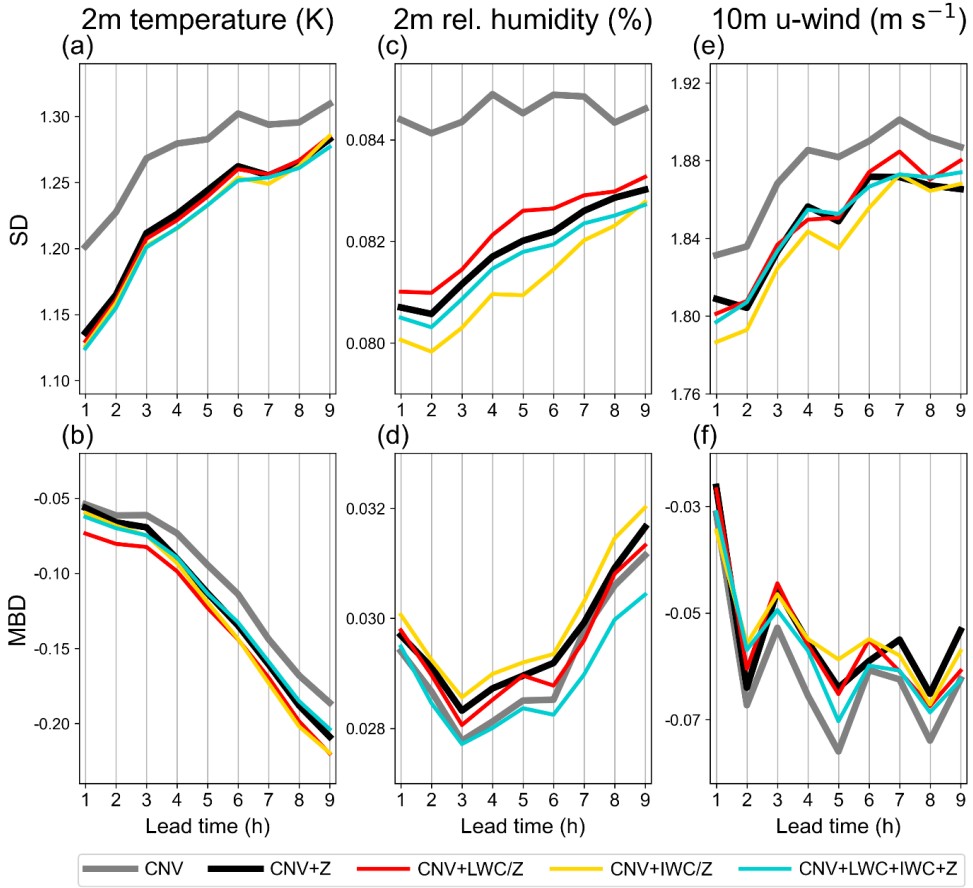

**Figure 10**: Mean standard deviations (SD; upper panel row) and mean bias deviations (MBD; lower panel row) of forecasted 2m temperature (left panel column), 2m relative humidity (middle panel column), and 10m u-wind (right panel column) from conventional observations in Germany as functions of the forecast lead time. Means are calculated over nine-hour forecasts initiated every third hour from hourly assimilation cycles with the assimilation configurations CNV (grey curves), CNV+Z (black curves), CNV+LWC/Z (red curves), CNV+IWC/Z (yellow curves), and CNV+LWC+IWC+Z (blue curves) using the found best DAP settings for the LWC and IWC assimilations (S2-06 and S1-02 in Table 2), and taking all rainfall cases C2017, S2017, and S2021 into account.





| DAP values | LH (km) | LV (ln(p)) | OE | LS (km) | LL | MV |
|---|---|---|---|---|---|---|
| Pre-selected (S-pre) | 16 | h.d. | 0.50 | 10 | -2.30 | 3 |
| Modification 1 | 8 | 0.2 | 0.25 | 5 | -1.15 | 25% |
| Modification 2 | 32 | 0.5 | 1.00 | 20 | -4.60 | 50% |

**Table 1**: Pre-selected and modified (modifications 1 and 2) values for the DAPs LH (horizontal observation localization length-scale in km), LV (vertical localization length-scale in logarithm of pressure ln(p)), OE (observation error standard deviation for log(LWC) and log(IWC)), LS (superobbing window size in km), LL (lower limit of the log(LWC) and log(IWC) data), and MV (minimum number of valid values for superobbing).

| DAP settings | LH (km) | LV (ln(p)) | OE | LS (km) | LL | MV |
|---|---|---|---|---|---|---|
| S1-01 | 16 | h.d. | 1.00 | 5 | -2.30 | 50 % |
| S1-02 | 8 | 0.5 | 0.25 | 10 | -1.15 | 50 % |
| S1-03 | 8 | 0.5 | 0.25 | 20 | -1.15 | 3 |
| S1-04 | 32 | 0.5 | 0.50 | 5 | -2.30 | 25 % |
| S1-05 | 8 | 0.2 | 0.25 | 10 | -4.60 | 50 % |
| S1-06 | 16 | h.d. | 0.50 | 20 | -1.15 | 25 % |
| S1-07 | 32 | 0.2 | 1.00 | 5 | -1.15 | 3 |
| S1-08 | 8 | 0.2 | 0.50 | 20 | -2.30 | 3 |
| S1-09 | 32 | 0.5 | 0.50 | 5 | -4.60 | 25 % |
| S1-10 | 16 | h.d. | 1.00 | 10 | -4.60 | 25 % |
| S1-11 | 32 | h.d. | 1.00 | 20 | -4.60 | 3 |
| S1-12 | 16 | 0.2 | 0.25 | 10 | -2.30 | 50 % |
| S2-01 | 16 | 0.2 | 1.00 | 20 | -1.15 | 50 % |
| S2-02 | 16 | 0.2 | 0.25 | 10 | -2.30 | 3 |
| S2-03 | 8 | h.d. | 1.00 | 20 | -1.15 | 3 |
| S2-04 | 16 | 0.2 | 1.00 | 20 | -2.30 | 50 % |
| S2-05 | 16 | h.d. | 0.25 | 10 | -2.30 | 50 % |
| S2-06 | 8 | 0.2 | 0.25 | 20 | -1.15 | 3 |
| S2-07 | 8 | 0.2 | 1.00 | 10 | -1.15 | 3 |
| S2-08 | 8 | h.d. | 0.25 | 10 | -1.15 | 50 % |
| S2-09 | 8 | h.d. | 1.00 | 20 | -2.30 | 50 % |
| S2-10 | 16 | h.d. | 0.25 | 10 | -2.30 | 3 |

**Table 2**: First and second near-random sample of DAP settings (S1-01 to S1-12 and S2-01 to S2-10) generated with Latin Hypercube Sampling from all the DAP values in Table 1 and with a reduced number of DAP values from Table 1 based on conisderation of the univariate measure $JQS_v$ (see Eq. (5) in Sect. 4.4) calculated with the first sample, respectively.