# Peer review of "Assimilation of 3D Polarimetric Microphysical Retrievals in a Convective-Scale NWP System"

_EGUsphere, 2023_

## Referee Comment (RC2)

**Specific comments**

The article investigates the impacts of assimilating retrievals of polarimetric radar data on the assimilation cycles as well as on skills of short-term forecasts, using the ICON-D2 KENDA system of DWD. Authors' efforts are greatly appreciated and it is very encouraging to see positive results of this study. However, I have several specific comments for authors to consider:

1. Language: 1) The language is not concise. For instance, at places "Assimilating LWC estimates instead of Z data where possible (CNV+LWC/Z)", "assimilation of IWC instead of Z where possible (CNV+IWC/Z)" and etc. would be enough to use the acronyms of experiments. 2) There are too many acronyms that seriously disturb the readability of the article since it is often difficult to tell what they are associated with, e.g., LC, LH, LV, OE, LS, LL, MV and ect. If they are really necessary (some of them seem not), could you make them more self-explaining or make a list for acronyms. Besides, it is uncommon to call "Data assimilation parameter". "Settings of the data assimilation system" may be more appropriate.

2. Assimilating of Z at the melting layer: 1) It should be noted that attenuation is normally switched on in EMVORADO, but the observations are corrected for attenuation. Authors should check if they are consistent. 2) I assume that the Mie scattering scheme is used here. As shown in Zeng et al. 2022a,b, the default paramertrizations for the melting layer in case of the Mie scheme result in unrealistically high reflectivities around the melting layer. Therefore, it is striking to assimilate Z only around the melting layer, which could be error-prone.

3. As shown in Zeng et al. 2016, Bick et al. 2016, the one produces smaller errors (likely due smaller localization radius, shorter update frequency and etc.) within assimilation cycles may not always win in forecasts, usually because of the imbalance issue that accelerates the error growth. Generally, it is inappropriate to say which one is the best only based on the performance in cycles but without looking at results of forecasts. Therefore, authors should avoid saying, e.g., "most successful assimilation settings", "finding the best DAP sets" and similar phrases. Maximally, it can be said that they result in smallest errors in terms of ...

4. Redundancy in appendix: It is unusual to describe the LETKF, FBI, FSS and BSS in full length since they are well-known. It would be enough to describe them shortly but provide their references.

**Minor comments**

1. Line 54: the error covariance matrix ==> the background error covariance matrix

2. Line 69: Bonavita et al. 2010 ==> Gastaldo et al. 2021

3. Line 115: EMVORADO may not stand for "Efficient Modular Volume Radar Forward Operator". Please ask Dr. Blahak Ulrich for this.

4. Line 160: reference for one-moment scheme

5. Line 175: Add the inflation method, RTPP and additive noise are applied for inflation.

6. Line 199: Please check if the used radar observations in 2021 are really of radial resolution of 0.25 km.

7. Line 236; The American climatological value 0.02 can be used for German radars?

8. Line 290: The superobbing described in Bick et al. 2016 is a bit outdated, the updated one is given in Zeng et al. 2021.

9. Line 304: better results than what?

10. Line 332: Please provide the frequency of boundary data

11. Line 350: Is the JQS a new metric introduced in the work? If yes, please explain more why this hybrid metric JQS is more interesting or important than FSS and BSS, otherwise, provide the reference.

12. Line 361: How are weights determined?

13. Line 387: Could authors be more specific about "which may be due to discrepancies between true and model microphysics"?

14. Line 485: Remove colors

15. Line 534: Strange phrase "first-guess precipitation forecasts" ==> first-guess of precipitation

**Reference**

1. Zeng, Yuefei, Yuxuan Feng, Alberto de Lozar, Klaus Stephan, Leonhard Scheck, Kobra Khosravianghadikolaei, and Ulrich Blahak. 2022b: Evaluating Latent-Heat-Nudging Schemes and Radar forward Operator Settings for a

Convective Summer Period over Germany Using the ICON-KENDA System, Remote Sensing, 14, 5295.

2. Zeng, Y. , H. Li, Y. Feng, U. Blahak, A. de Lozar, J. Luo, J. Min, 2022a: Study of Sensitivity of Observation Error Statistics of Doppler Radars to the Radar forward Operator in Convective-Scale Data Assimilation. Remote Sensing, 14, 3685.

3. Zeng, Y., T. Janjic, Y. Feng, U. Blahak, A. de Lozar, E. Bauernschubert, K. Stephan, J. Min, 2021: Interpreting estimated Observation Error Statistics of Weather Radar Measurements using the ICON-LAM-KENDA System, Atmospheric Measurement Techniques, 14, 5735–5756.

---

## Author Comment (AC1)

**Reply to reviewer #1**

**General comments:**

This manuscript is a novel attempt to assimilate C-band polarimetric microphysical retrievals into a convection-resolving model in Germany compared to the legacy assimilation approach using Z. The authors find generally positive benefits, particularly for LWC, with IWC neutral/beneficial in stratiform cases but with harmful effects in a convective case. This is not particularly surprising, as the retrievals used are not formulated for rimed ice/hail, but it is nonetheless a good demonstration of what can happen if the retrieval equations are not applied appropriately/selectively to how they were formulated. The results overall indicate that the addition of LWC and IWC retrievals, with Z, results in the best forecasts, and encourages further exploration. The manuscript is very well written, with a thorough introduction section that clearly establishes the state-of-the-art of polarimetric radar DA and its challenges. The basis and results of the study are novel and timely as polarimetric radars are increasingly implemented and their information content explored. There are a number of improvements I'd like to see to the manuscript primarily regarding clarity or requests for additional context/information. While I don't think it is necessary for publication, I did feel the manuscript could benefit from one or two plots actually depicting the results of the different assimilation experiments (e.g., actual post-assimilation hydrometeor and/or moisture fields, or QPF fields, vs. the observed QPE; the distribution/statistics of the IWC/LWC fields, etc.), so that readers can get a visual sense of the effects these are having rather than solely relying on only statistics. Despite this, I believe the manuscript will be ready for publication once the following minor comments are addressed.

*Dear reviewer, thank you very much for the positive feedback and encouraging words. We appreciate your efforts and your suggestion to include visual impressions of the effect of the LWC/IWC assimilation. We indeed already tested the visualization of e.g. QPF vs. QPE fields when LWC/IWC assimilation was included compared to the case when only Z was assimilated. However, while some slight positive impact could be noted in terms of FSS, BSS and FBI, it was very difficult to identify the differences visually. Figure A shows an example for a single observation experiment we made. While some effect of single Z or LWC/IWC observations on the QPF fields can be noted, the differences in the fields between the Z and LWC/IWC assimilation is difficult to identify visually. Therefore, we decided to not visualize the QPF/QPE fields in the paper and to just focus on the statistics of the FSS, BSS and FBI measures. Still, a deeper look into the impact of the LWC/IWC assimilation on the spatial statistics of the LWC, IWC or moisture fields could be worth looking at in the future. Thanks again for your helpful comments and suggestions!*

*Note that all new line numbers in the following refer to the manuscript file without tracked changes!*

[Figure]

*Figure A: a) 1-hour RADOLAN precipitation accumulation for 14 July 2021 16 UTC, d) first guess (FG) 1-hour precipitation for the same hour without data assimilation, b) difference to d) for assimilation of a single Z super observation at the position below the melting layer marked by the black diamond, c) same as c) but with a single LWC super observation, e) and f) are as b) and c) but for a superobbing point above the melting layer marked by the black pentagons.*

**Specific comments:**

1. L26: Please define FSS and FBI for the abstract.

   *Of course, thank you! (L26,27)*

2. L103: It may be worth mentioning that beyond the rudimentary treatment of PSDs, hydrometeor shapes and orientations are rarely (if ever?) taken into account at all within model microphysics schemes.

   *Thank you for that comment. We adjusted the text accordingly. (L104)*

3. L160: Which microphysics scheme is used in the ICON-D2? A citation is needed here.

   *We included the corresponding citation (Doms et al., 2011). (L161)*

4. L180: In this study, when Z is assimilated (either in CNV+Z, CNV+[LWC + IWC]/Z, or CNV + LWC + IWC + Z) does this mean it is both directly assimilated *and* assimilated through LHN? It is not explicitly clear to me how Z is being used as the reference assimilation study. (Edit: I see now on L338 it is stated that LHN is excluded in this case, but this wasn't clear at first when mentioned in the context of the operational Z assimilation scheme).

*We included the sentence "Note that LHN and the assimilation of 3D V observations are not applied in this study (see below)." for clarity. (L183,184)*

5. L204 and elsewhere: Is there a specific reason the variables are included as ZDR, PHIDP, KDP, and RHOHV rather than $Z_{DR}$, $\Phi_{DP}$, $K_{dp}$, and $\rho_{hv}$? It isn't critical, but I think use of the variables are more standard notation, would clean up the equations/discussion, and not suggest that ZDR, RHOHV, PHIDP, and KDP might be acronyms to unfamiliar readers.

   *This is a good point. There is no real reason why we wrote the variables like that. We changed the text to the standard notation everywhere. (e.g. L206-208)*

6. L206: Strictly speaking, isotropic scatterers will have an intrinsic ZDR of exactly 0 dB rather than one close to 0 dB.

   *Of course! (L209)*

7. L217: This value of $\rho_{hv}$ seems much lower than what I would have expected. At S band, $\rho_{hv}$ in dry snow/ice (rather than melting particles, which are neglected in these retrievals) is usually nowhere near 0.85 unless it is very large hail experiencing resonance scattering, in which case the use of the IWC retrieval equations would be inappropriate anyway (both due to it being hail rather than snow and due to the non-Rayleigh scattering). With all the tests done to find the optimal DAP, were any tests done to examine the impact of these thresholds and find optimal values? What did the typical distribution of $\rho_{hv}$ values actually look like above the ML? I am concerned that using such a low $\rho_{hv}$ threshold aloft will necessarily retain data that is going to have very noisy $\Phi_{DP}/K_{dp}$ and thus noisy/erroneous IWC retrievals that should not be assimilated. I would be curious if a much more stringent $\rho_h$ threshold (perhaps something like 0.92 or greater) would result in better retrieval/assimilation results.

   *This is a legal comment. We chose the RHOHV threshold just based on the value listed in the Kumjian (2013a) paper for S-band. Yes, possibly this too low threshold contributes to the rather worse assimilation results for IWC by retaining noisy data. However, we did not test the sensitivity of the assimilation experiments to the RHOHV thresholds. This would be worth an investigation in the future. Besides, you find a typical distribution of RHOHV and Z in our data in Figures B and C.*

[Figure]

*Figure B: QVP of RHOHV of radar UMD at 5.5 degrees elevation for the stratiform day 25 July 2017.*

[Figure]

*Figure C: Corresponding QVP of Z (in dBZ).*

8. L221: How is the bottom of the melting layer determined, as shown in Fig. 2? I assume some sort of $\rho_{hv}$ threshold when using a QVP, but for the convective case (for which I presume model data was relied upon) it is less clear.

*We used QVPs of RHOHV, determined the approximate center of the melting layer within 6-hour intervals visually and subtracted/added 500m to identify the approximate upper and lower boundaries. In convective situations, we used the closest operational radio sounding to approximate the height of the center (approx.. 0°C isotherm) of the melting layer and then added/subtracted 500m to obtain the upper/lower boundaries. We changed the text to: "The height of the melting layer is determined from so-called Quasi-Vertical Profiles (i.e., azimuthal medians of PPIs measured at sufficiently high*

*elevations and transferred to range-height displays; Trömel et al., 2014; Ryzhkov et al., 2016), as derived from PPIs measured at a 5.5 degree elevation angle, or from the nearest operational DWD radio sounding, especially in convective situations." (L227)*

9. L225: I appreciate why such a large window is needed for consistency, but depending on how small and isolated the convection is, could 9 km be too large a window and end up heavily incorporating data boundary regions into the Kdp calculation for precipitation regions?

   *Yes, this is a big problem with the low radial resolution of most of the radar data used. On the one hand side, a small window size close to strong cells is desireable to keep the small scale features, but on the other hand side, the low resolution may introduce errors when using a small window (because only little data points contribute to the estimates). The newer radar data of DWD (since march 2021) has a 4 times higher radial resolution (250m) compared to the older data (1 km). Thus, it would be worth exploring how an adjustment of the window size for KDP depending on the precipitation situation may change the assimilation results. Unfortunately, the convective case considered is from 2017 and thus has the low 1-km resolution. For sure we will keep on working on the topic including the relevance of the enhanced radial resolution and hopefully the assimilation of polarimetric microphysical retrievals will become operational at DWD in the future. However, this is beyond the scope of this study.*

10. L246 and elsewhere: I assume log here is $\log_{10}$ and not ln? It may be helpful just to clarify.

    *Yes, this is a bit unclear. We changed "log" to "log10" everywhere. (e.g. L249)*

11. L256: It may also be worth mentioning in addition to just hail that R(A) at C band struggles from the resonance scattering of medium-large-sized raindrops, which causes R(Kdp) to outperform R(A) for moderate to heavy rainrates.

    *Thanks for the comment. We added one sentence: "In addition, resonance scattering of medium and large sized raindrops at C-band may favour the use of LWC(KDP) compared to LWC(A) in moderate to heavy rain." (L259-261)*

12. L207: I am a bit confused by this. The Figure 2 caption says LS = 10 km, which would make it equivalent to LC, but here it makes it sound like LS != LC.

    *We hope it becomes more clear by adding the word "also": "The number of radar bins contributing to the averaging decreases with increasing distance from the radar, and the window size for the averaging (winsize_avg in km) is equal to res_cartesian in KENDA, but is also modified in our study while keeping res_cartesian constant." (L293) Note that the acronyms "LC" and "LS" are changed here to "res_cartesian" and "winsize_avg", respectively, for reasons of clarity.*

13. L305: While the results of these tests are not shown here, previous studies such as Liu et al. (2020) have demonstrated the same thing for hydrometeor mixing ratios and could be cited here, if desired.

Liu, C., M. Xue, and R. Kong, 2020: Direct variational assimilation of radar reflectivity and radial velocity data: Issues with nonlinear reflectivity operator and solutions. Mon. Wea. Rev. **148**.

   *Thanks for the suggestion. We added the citation! (L311)*

14. L331: I may be misunderstanding something simple, but if the assimilation process during this period is what is done operationally (CONV + Z + LHN), what is the reason the data was obtained 24-35 hours beforehand and further spun up rather than just using the operational data from the model initial time (e.g, 00 UTC 13 July 2021 instead of 00 UTC 12 July 2021)?

*Good point. The problem is a very limited access to the DWD data. The model data was only provided for the times 00 UTC 12 July 2013 etc. while the time interval of interest only started 24 etc. hours later. To reduce confusions, we rephrased that the model data was provided for the initial times of the experiment periods, which is reasonable because the assimilation cycles used to obtain the model data at the starting times of the experiments are identical to the operational cycles. We rewrote the paragraph 4.3 accordingly. (L334-338)*

15. L336: I am curious about the assimilation of Z data only within the melting layer, where the relationship between Z and the microphysical state variables becomes most complicated and obfuscated. If anything the data in the melting layer is often neglected because it can result in some large errors.

*That is true. However, in the operational ICON-D2 routine at DWD, Z is assimilated within the melting layer. Since the operational Z assimilation (without LWC or IWC) is used as reference in this paper, the configurations assimilating LWC/IWC also need to include the Z assimilation in the melting layer for comparability. However, as indicated by the reviewer, results of assimilating Z data in the melting layer highly depend on the operator's melting scheme. Different flavours of the Maxwell-Garnett-, Bruggemann- and Wiener Effective Medium Approximations (EMA) can be chosen to explore the uncertainty in the melting layer. We are currently investigating deficiencies of the simulated melting layer signature, parts of which may be attributed to the fact that in the model microphysics parameterization, meltwater from snow, graupel, and ice is instantaneously shedded into the rain class, causing too small and too few remaining frozen particles in the melting layer. The current version of EMVORADO estimates a melted fraction as function of temperature and particle size (Blahak 2016) as part of the remaining frozen mass without "back-shuffling" some rain water to the particles. This leads to a systematic underestimation of the melting effect in all radar moments, despite the quite detailed consideration of various EMAs for the effective refractive index. We will explore better approaches, e.g. a wet snow class borrowing parts of the rain and snow mixing ratios and mix them (e.g. Jung et al. 2008a, 2008b, 2010), to reduce the bias to the observations. However, as soon as model microphysics with explicit mixed-phase snow, graupel, and hail (e.g, as in Frick et al. 2013 for snowflakes) becomes available in the future, their liquid fraction could be directly applied in the forward operator.*

*Blahak, U.: RADAR_MIE_LM and RADAR_MIELIB – Calculation of Radar Reflectivity from Model Output, COSMO Technical Report 28, Consortium for Small Scale Modeling (COSMO), available at: http://www.cosmo-model.org/content/model/documentation/techReports/cosmo/docs/techReport28.pdf (last access: 10 January 2022), 2016. a, b, c, d, e*

*Jung, Y., G. Zhang, and M. Xue, 2008a: Assimilation of simulated polarimetric radar data for a convective storm using the ensemble Kalman filter. Part I: observation operators for reflectivity and polarimetric variables. Monthly Weather Review, 136 (6), 2228-2245, DOI: 10.1175/2007MWR2083.1.*

*Jung, Y., M. Xue, G. Zhang, and J. M. Straka, 2008b: Assimilation of simulated polarimetric radar data for a convective storm using the ensemble Kalman filter. Part II:*

*impact of polarimetric data on storm analysis. Monthly Weather Review, 136 (6), 2246-2260, DOI: 10.1175/2007MWR2288.1.*

*Jung, Y., M. Xue, and G. Zhang, 2010: Simulations of polarimetric radar signatures of a supercell storm using a two-moment bulk microphysics scheme. Journal of Applied Meteorology and Climatology, 49 (1), 146-163, DOI: 10.1175/2009JAMC2178.1.*

*Frick, C., Seifert, A., and Wernli, H.: A bulk parametrization of melting snowflakes with explicit liquid water fraction for the COSMO model, Geosci. Model Dev., 6, 1925–1939, https://doi.org/10.5194/gmd-6-1925-2013, 2013.*

16. L512: I apologize if I'm misunderstanding, but these two sentences seem contradictory to me. It is stated that the best forecasts are achieved when there is limited influence from the radar-based retrievals (I assume in terms of impact to the analysis by way of a larger lower threshold, rather than spatial localization?). But subsequently it is stated that a smaller observation error standard deviation, which I believe would enhance the weighting toward observations, is also beneficial.

    *Indeed, this may be confusing. We changed the first sentence as follows: "Thus, best first guess of precipitation forecasts are achieved when the influence of the observed microphysical estimates on the model state is rather small in terms of observation localization length-scale and lower data limit. A rather small observation error standard deviation of 0.25 in log10(LWC) and log10(IWC) was most successful." (L523-527)*

17. L548: This, to me, is an absolutely crucial piece of information for interpreting the results of assimilating LWC/IWC in this study and needs to be discussed earlier in the paper in the assimilation section. While I think the adjustment of the moisture variables (rather than precipitation variables) is in general an important aspect of radar data assimilation, I also feel the impacts of assimilating a "bad" retrieval (say above the ML in what is actually graupel/hail) on moisture may be even more deleterious than if it were acting on qs, which could precipitate out relatively quickly and lead to little harm to the forecast. This seems like an important follow-up study since it is likely not what readers would have expected at first.

    *This is a very interesting point. Following this idea, producing increments also in qs when assimilating IWC would potentially not change the results much. This definitely requires future investigation. We included one sentence in section 4.2 in order to enable a better interpretation of the results: "Moreover, microphysical analysis increments of only cloud water mixing ratio and specific humidity are produced, i.e., not all available hydrometeor species (e.g., rain, cloud ice, and graupel mixing ratios) are updated individually in KENDA's standard configuration." (L.320-322)*

18. L565: Could additional studies be done that only assimilate the IWC retrievals in areas identified as snow in convective cases, thus potentially limiting the presumed harmful impacts of assimilating poor/inappropriate retrievals of IWC? With benefits coming from Z alone, I am curious if it could still be wise to assimilate IWC only selectively, in addition to LWC and Z (and V and CNV).

    *Thank you for this input! A hydrometeor classification prior to assimilation may have some potential to eliminate the harmful effects of hail/graupel in convective precipitation. Also, a selective assimilation of IWC in such situations could be worth being tested.*

**Technical corrections**:

1. L31: "adequat" should be "adequate"
2. L330: "and including LHN" should be "including LHN"
3. L538: "adjusted" should be "developed"

   *Points 1) and 3) were corrected (L31 and L541), point 2) was not corrected because the respective sentence was deleted. Thank you again for your time!*

---

## Author Comment (AC2)

**Reply to reviewer #2**

**Specific comments**

The article investigates the impacts of assimilating retrievals of polarimetric radar data on the assimilation cycles as well as on skills of short-term forecasts, using the ICON-D2 KENDA system of DWD. Authors' efforts are greatly appreciated and it is very encouraging to see positive results of this study. However, I have several specific comments for authors to consider:

1. Language: 1) The language is not concise. For instance, at places "Assimilating LWC estimates instead of Z data where possible (CNV+LWC/Z)", "assimilation of IWC instead of Z where possible (CNV+IWC/Z)" and etc. would be enough to use the acronyms of experiments. 2) There are too many acronyms that seriously disturb the readability of the article since it is often difficult to tell what they are associated with, e.g., LC, LH, LV, OE, LS, LL, MV and ect. If they are really necessary (some of them seem not), could you make them more self-explaining or make a list for acronyms. Besides, it is uncommon to call "Data assimilation parameter". "Settings of the data assimilation system" may be more appropriate.

   *Dear reviewer, thank you very much for your time and helpful comments.*

   *Note that all new line numbers in the following refer to the manuscript file without tracked changes!*

   *1: Yes, we first tried to explain the assimilation configurations like "Assimilating LWC estimates instead of Z data where possible (CNV+LWC/Z)" repeatedly in the text to remind the reader, but we followed now your advice in the revised manuscript. (e.g. L396,397)*

   *2: Yes, we made the acronyms more self-explaining throughout the document: LC -> res_cartesian; LS -> winsize_avg; LH -> obsloc_hor; LV -> obsloc_ver; OE -> obserr_std; LL -> lower_lim; MV -> minnum_vals (see e.g. L292, 294, 297). Moreover, we use the wording "Data assimilation parameter" for winsize_avg, obsloc_hor etc. to clearly distinguish it from the data set configurations like CNV, CNV+LWC/Z etc., which also belong to the "settings of the data assimilation system". Therefore, we would like to keep the acronym "DAP" in our study.*

2. Assimilating of Z at the melting layer: 1) It should be noted that attenuation is normally switched on in EMVORADO, but the observations are corrected for attenuation. Authors should check if they are consistent. 2) I assume that the Mie scattering scheme is used here. As shown in Zeng et al. 2022a,b, the default parametrizations for the melting layer in case of the Mie scheme result in unrealistically high reflectivities around the melting layer. Therefore, it is striking to assimilate Z only around the melting layer, which could be error-prone.

   *1: Thank you for that important comment. Yes, it is true that attenuation of Z is turned on in ICON-EMVORADO by default. We talked to Ulrich Blahak from DWD about that topic. In his research group they are still elaborating on finding a way to compare observed attenuation-corrected Z with EMVORADO-simulated non-attenuated Z. However, at the moment Z assimilation works best when attenuation is switched on in EMVORADO while assimilating attenuation-corrected observed Z. This may be caused by the point that the ICON-EMVORADO tends to overestimate Z on average which may partly be compensated for by the applied attenuation. A brief view into the data showed a general consistency of the simulated and observed Z fields. Since this study assimilates Z as performed in the operational routine, attenuation-corrected observed Z and attenuated simulated Z are used. Besides the attenuation effect, inconsistencies between modelled and observed Z in*

*the ICON-EMVORADO have become evident e.g. in rain below the melting layer because of too large simulated drops, and around the melting layer because of excessive graupel production. These are important points to be addressed in the future but would by far go beyond the scope of this study.*

*2: Yes, Mie scattering is used. In the operational ICON-D2 routine at DWD, Z is assimilated within the melting layer. Since the operational Z assimilation (without LWC or IWC) is used as reference in this paper, the configurations assimilating LWC/IWC also need to include the Z assimilation in the melting layer for comparability. However, results of assimilating Z data in the melting layer highly depend on the operator's melting scheme. Different flavours of the Maxwell-Garnett-, Bruggemann- and Wiener Effective Medium Approximations (EMA) can be chosen to explore the uncertainty in the melting layer. We are currently investigating deficiencies of the simulated melting layer signature, parts of which may be attributed to the fact that in the model microphysics parameterization, meltwater from snow, graupel, and ice is instantaneously shedded into the rain class, causing too small and too few remaining frozen particles in the melting layer. The current version of EMVORADO estimates a melted fraction as function of temperature and particle size (Blahak 2016) as part of the remaining frozen mass without "back-shuffling" some rain water to the particles. This leads to a systematic underestimation of the melting effect in all radar moments, despite the quite detailed consideration of various EMAs for the effective refractive index. We will explore better approaches, e.g. a wet snow class borrowing parts of the rain and snow mixing ratios and mix them (e.g. Jung et al. 2008a, 2008b, 2010), to reduce the bias to the observations. However, as soon as model microphysics with explicit mixed-phase snow, graupel, and hail (e.g, as in Frick et al. 2013 for snowflakes) becomes available in the future, their liquid fraction could be directly applied in the forward operator.*

*Blahak, U.: RADAR_MIE_LM and RADAR_MIELIB – Calculation of Radar Reflectivity from Model Output, COSMO Technical Report 28, Consortium for Small Scale Modeling (COSMO), available at: http://www.cosmo-model.org/content/model/documentation/techReports/cosmo/docs/techReport28.pdf (last access: 10 January 2022), 2016.  a, b, c, d, e*

*Jung, Y., G. Zhang, and M. Xue, 2008a: Assimilation of simulated polarimetric radar data for a convective storm using the ensemble Kalman filter. Part I: observation operators for reflectivity and polarimetric variables. Monthly Weather Review, 136 (6), 2228-2245, DOI: 10.1175/2007MWR2083.1.*

*Jung, Y., M. Xue, G. Zhang, and J. M. Straka, 2008b: Assimilation of simulated polarimetric radar data for a convective storm using the ensemble Kalman filter. Part II: impact of polarimetric data on storm analysis. Monthly Weather Review, 136 (6), 2246-2260, DOI: 10.1175/2007MWR2288.1.*

*Jung, Y., M. Xue, and G. Zhang, 2010: Simulations of polarimetric radar signatures of a supercell storm using a two-moment bulk microphysics scheme. Journal of Applied Meteorology and Climatology, 49 (1), 146-163, DOI: 10.1175/2009JAMC2178.1.*

*Frick, C., Seifert, A., and Wernli, H.: A bulk parametrization of melting snowflakes with explicit liquid water fraction for the COSMO model, Geosci. Model Dev., 6, 1925–1939, https://doi.org/10.5194/gmd-6-1925-2013, 2013.*

3.  As shown in Zeng et al. 2016, Bick et al. 2016, the one produces smaller errors (likely due smaller localization radius, shorter update frequency and etc.) within assimilation cycles may not always win in forecasts, usually because of the imbalance issue that accelerates the error growth. Generally, it is inappropriate to say which one is the best only based on the performance in cycles but without looking at results of forecasts. Therefore,

authors should avoid saying, e.g., "most successful assimilation settings", "finding the best DAP sets" and similar phrases. Maximally, it can be said that they result in smallest errors in terms of ...

*You are absolutely right. An "optimal" or "best" DA setting should refer to the quality of the forecast and not on the quality of first-guesses in assimilation cycles. Therefore, we changed the formulation at all positions in the document. (e.g. L405-407)*

4. Redundancy in appendix: It is unusual to describe the LETKF, FBI, FSS and BSS in full length since they are well-known. It would be enough to describe them shortly but provide their references.

*We appreciate your suggestion and removed the appendix. In the document text the references to LETKF, FBI, FSS, and BSS are cited and should be sufficient for understanding.*

**Minor comments**

1. Line 54: the error covariance matrix ==> the background error covariance matrix

   Thanks! (L55)

2. Line 69: Bonavita et al. 2010 ==> Gastaldo et al. 2021

   Thanks! (L71)

3. Line 115: EMVORADO may not stand for "Efficient Modular Volume Radar Forward Operator". Please ask Dr. Blahak Ulrich for this.

   Yes, it should be "Efficient Modular VOlume scan RADar Operator". Thank you! (L116,117)

4. Line 160: reference for one-moment scheme

   Citation now included: Doms et al., 2011 (L161)

5. Line 175: Add the inflation method, RTPP and additive noise are applied for inflation.

   We included one sentence. (L175-177)

6. Line 199: Please check if the used radar observations in 2021 are really of radial resolution of 0.25 km.

   Yes, they have a resolution of 0.25 km while the 2017 cases have a 1 km resolution.

7. Line 236; The American climatological value 0.02 can be used for German radars?

   As there is no comparable value for central Europe it may be used approximatively.

8. Line 290: The superobbing described in Bick et al. 2016 is a bit outdated, the updated one is given in Zeng et al. 2021.

   We included the newer reference. (L296)

9. Line 304: better results than what?

   We clarified the point. (L309-311)

10. Line 332: Please provide the frequency of boundary data

    We included the frequency of one hour. (L338)

11. Line 350: Is the JQS a new metric introduced in the work? If yes, please explain more why this hybrid metric JQS is more interesting or important than FSS and BSS, otherwise, provide the reference.

*Yes, this metric is new. We modified the respective sentence: "The results of using the DAP configurations/values are compared with each other in terms of both first-guess deterministic and ensemble QPF quality via a single univariate measure newly introduced here – the joint quality score (JQS).." (L355-360); "While changes in deterministic and ensemble QPF quality with respect to the CNV+Z configuration are not always consistent, the JQS provides a useful measure for the overall intercomparison of DA settings.." (L361-363)*

12. Line 361: How are weights determined?

"weights are determined by the fractions of threshold exceedances for a given time and threshold of the total number of exceedances at all thresholds (0.5, 1.0, 2.0, and 4.0 mm h-1) and events (C2017, S2017, and S2021) in the RADOLAN data (see Fig. 3)" (L371-374)

13. Line 387: Could authors be more specific about "which may be due to discrepancies between true and model microphysics"?

*We removed the sentence about the discrepancies. (L400)*

14. Line 485: Remove colors

*We removed the colors. (L498)*

15. Line 534: Strange phrase "first-guess precipitation forecasts" ==> firstguess of precipitation

*We modified it. (L523)*